# Influence of Snow Spatial Variability on Cosmic Ray Neutron SWE: Case Study in a Northern Prairie

Haejo Kim[1], Eric Sproles[2,3], Samuel E. Tuttle[1]

[1]Department of Earth & Environmental Sciences, Syracuse University, Syracuse, NY, 13244, USA
[2] Department of Earth Sciences, Montana State University, Bozeman, MT, 59717, USA
[3] Geospatial Core Facility, Montana State University

*Correspondence to*: Haejo Kim (hkim139@syr.edu)

**Abstract.** Monitoring prairie snow is difficult due to its extreme spatial variability from windy
conditions, gentle topography, and low tree cover. Previous work has shown that a noninvasive, aboveground Cosmic Ray Neutron Sensor (CRNS) placed at the Central Agricultural Research Center (CARC; 47.07º N, 109.95º W), an agricultural research site within a semi-arid prairie environment managed by Montana State University, was sensitive to both the low snow amounts and spatial variability of prairie snow. In this study, we build upon previous work to understand how different snow
distributions would have influenced CRNS measurements at the CARC. Specifically, we compared the changes in neutron counts and snow water equivalent (SWE) after relocating our CRNS probe at the CARC using the Ultra Rapid Neutron-Only Simulation (URANOS) and comparing them to uniform snow distributions. Neutron counts from simulations with a shallow, heterogeneous snowpack were higher compared to neutron counts from simulations with a uniform snowpack. While areas of higher
snow accumulation reduced neutron counts, the low SWE made it difficult to discern a consistent relationship between SWE and neutron counts. Despite this, our analysis indicates that a naive CRNS placement was 2 to 5 times more likely to yield representative SWE estimates compared to a similarly placed snow scale. CRNS showed better agreement with lidar-derived SWE at our prairie site compared to several gridded snow products. We show CRNS can provide valuable information about shallow,
heterogeneous snowpacks in prairie and other environments and can benefit future missions from UAV and satellite platforms.

## 1 Introduction

Seasonal snow plays an important hydrologic and climatic role in the Earth system. Seasonal snow covers an average of 31% of the Earth's surface annually (Tsang et al., 2022). A major component of
the Western United States' water supply originates from seasonal snowpack, feeding the needs of over 60 million people (Bales et al., 2006). Prairie snow can make up to 25% of the global snow cover (Sturm and Liston, 2021). Mid-latitude semi-arid prairie environments, such as those found in the interior Great Plains of North America (i.e. northern states such as Montana and extending north into Canada) are dependent on snow. Over 80 to 85% of streamflow in the Northern Great Plains originates

from snow (Gray, 1970), despite accounting for 20% of the annual precipitation (Aase and Siddoway, 1980).

       Snow cover in the prairie is known for its extreme spatial heterogeneity, mainly due to strong surface winds, gentle topography, and spatial variability in vegetation (Gray, 1970). Figure 1 depicts the variability that snow can exhibit in a prairie environment. Strong winds in an open, flat expanse of land
scours snow, causing wind erosion, enhancing sublimation, and transporting 75% of the annual snowfall (Gray, 1970; Harder et al., 2019). The effects of blowing snow are affected by changes in surface roughness such as vegetation which allows for preferential deposition and accumulation of snow along natural barriers (Harder et al., 2019; Kort et al., 2012). These areas of preferential deposition can build snow drifts as shown in Fig. 1a that can grow over 1 m tall and can transition to bare ground over a
spatial scale of meters to tens of meters.

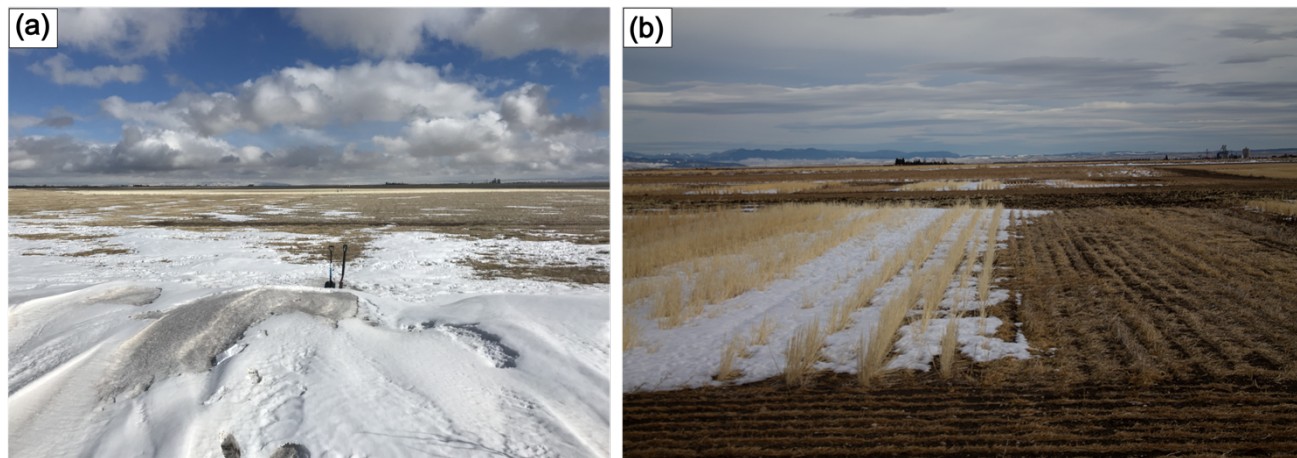

**Figure 1 Field images depicting the heterogeneity of snow in a prairie environment from winter 2020-2021. (a) Image taken on top of > 1 m snow drift, looking east, with snow disappearing as you move away from the snow drift. (b) Standing crop stubble is used**
**to trap snow for early spring melt. Field images were provided by Dr. Eric Sproles.**

       Figure 1b shows how vegetation variation due to agriculture in the Northern Great Plains can drive preferential snow accumulation. The introduction of dryland cropping techniques, such as no till (or zero tilling) allows certain winter wheat crops to grow in the Northern Great Plains changing the surface roughness of the prairies (Nielsen et al., 2005; Aase and Siddoway, 1980; Harder et al., 2019).
The increased surface roughness from crops allow for preferential deposition of snow, reducing the blowing snow process (Harder et al., 2019). In addition, farmers can leave standing crop stubbles, like in Fig. 1b, to aid in trapping snow and reducing snow erosion in order provide water recharge and manage infiltration and runoff (Aase and Siddoway, 1980; Harder et al., 2019). Due to the semi-arid climate in the Northern Great Plains, water use must be efficient for agricultural fields to be productive.
Thus, agricultural development in the prairies has increased the need to capture snow for early season melt water. Accurate SWE measurements in prairie environments are thus relevant for maximizing agricultural water use efficiency.

       Snow heterogeneity introduces an important question in water resources management: How and where can we effectively measure snow water equivalent (SWE) in prairies and other similar

environments? Traditional manual snow measurements from snow pits are labor-intensive and are best applied in deep snow. In prairie environments, snow pit measurements of snow density are usually restricted to snow drifts and are difficult to collect in shallower prairie snowpack. In addition, continuous SWE monitoring through snow pillows or snow scales like those found in the snow telemetry (SNOTEL) network from the US Department of Agriculture Natural Resources Conservation Service (USDA NRCS) (Serreze et al., 1999), are not as effective in the prairie due to wind erosion and transport. Additionally, Fig. 1 shows how the placement of a snow pillow or snow scale (e.g. in an area that accumulates a snow drift or an area that is wind-scoured) could result in very different snow measurements, some (or all) of which may not reflect the areal average SWE. Another alternative is to measure SWE at larger scales through remote sensing on satellite or airborne platforms. However, satellite and airborne remote sensing of SWE in the Northern Great Plains is currently limited by the SWE variability at the subpixel scale (Tuttle et al., 2018).

To overcome these limitations in snow observations in the prairies, we installed a Cosmic Ray Neutron Sensor (CRNS) to measure the SWE at an agricultural research site in the plains of central Montana. CRNS instruments detect the background neutron flux that is generated when cosmic rays interact with matter on Earth (Desilets et al., 2010). Neutrons are extremely sensitive to hydrogen, which can either be absorbed if the neutron is thermalized or slowed down due to energy loss from elastic collisions with hydrogen atoms (Zreda et al., 2012). Thus, a CRNS detector measures these attenuated neutrons, which is inversely related to the amount of hydrogen atoms in its immediate surroundings. The most common source of hydrogen in the environment is water molecules in the atmosphere (Rosolem et al., 2013; Zreda et al., 2012), vegetation (Baroni et al., 2018; Franz et al., 2015), and soils (e.g., lattice water and organic matter) (Bogena et al., 2013; Franz et al., 2013). After accounting for all other hydrogen pools, CRNS estimates of soil moisture and SWE are made over an approximate operational radius of 150 to 250 m (for aboveground CRNS) by detecting the neutron flux over time (Zreda et al., 2008; Royer et al., 2021). The non-invasive and large footprint of CRNS has intriguing potential to overcome the issues of traditional continuous snow monitoring in heterogeneous shallow to moderate snowpacks. It also helps to mitigate a common issue in hydrology: bridging the scale gap between point measurements and areal measurements, such as remote sensing or modelling studies, by providing measurements of areal SWE at an intermediate or similar spatial resolution (Blöschl, 1999; Iwema et al., 2015; Schattan et al., 2020).

Previous research has shown that CRNS estimates of SWE at an agricultural prairie site in central Montana agree with spatially weighted digital snow models (DSMs) from UAV light detection and ranging (lidar) flights and modeled CRNS estimates, despite extreme spatial heterogeneity of the snowpack surrounding the detector (Woodley et al., 2024). CRNS has been noted to be sensitive to bare ground patches, usually increasing the neutron counts (Schattan et al., 2019). We build on our previous research from Woodley et al. (2024) to analyze the effects of snow heterogeneity within the operational footprint of the CRNS using neutron transport modelling. From these results, we provide insights and guidelines on best practices to site future CRNS probes with respect to shallow, heterogenous snowpacks. We also use a synthetic analysis to compare the reliability of a naïve CRNS placement in a shallow, heterogeneous snowpack against a similarly sited snow scale. Finally, we compare CRNS estimates and currently available gridded SWE products to lidar- and ground-based SWE measurements

and find that CRNS measurements can be a reliable ground truth for remote sensing applications in the prairies.

## 2 Study Area

The modelling domain for this study is a 1 km$^2$ region of the Central Agricultural Research Center
(CARC), an agricultural research site managed by Montana State University, located in central Montana (47.057510° N, 109.952945° W; see Fig. 2). The CARC hosts ongoing agricultural research where researchers investigate different crop varieties, cropping strategies, and soil biogeochemistry. Crops typically grown at the CARC include cereals, grasses, legumes, and broadleaf plants. Some crops persist into the winter as stubble at the CARC, depending on harvest practices (Palomaki and Sproles,
2023). The elevation of the study region ranges from 1287 m to 1298 m. Soils at the CARC are primarily well-drained, shallow clay loams (Palomaki and Sproles, 2023). We observed average air temperatures of -0.4°C (-3.7°C during December-February (DJF)), average air pressure of 870 mb, and average relative humidity of 62.8% throughout the winter of 2020-2021 (November through April). A CRNS (CRS2000/B from HydroInnova LLC, Albuquerque, NM, USA) was deployed at the site in the
winter of 2020/2021, coincident with the SnowEx 2021 Prairie field campaign, to measure the low-energy cosmic ray-induced neutrons (Woodley et al., 2024).

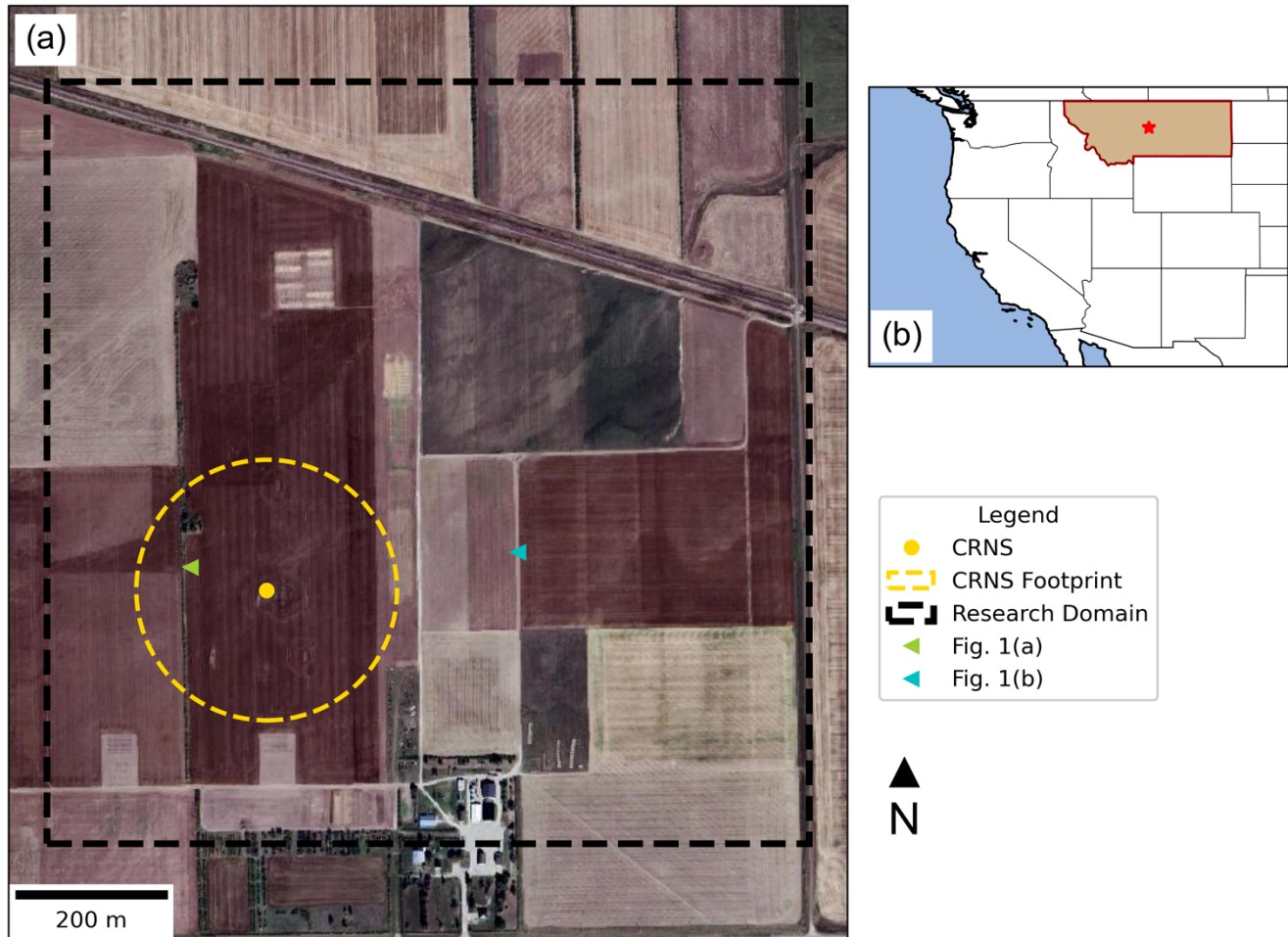

**Figure 2 Basemap of study site. (a) The 1 km² research domain outlined by the dashed black box at the Central Research Agricultural Center (CARC). The CRNS location is marked by the yellow dot and the estimated 171 m footprint (calculated in Woodley et al., 2024) is shown in the dashed yellow circle. The approximate locations where Fig. 1a (green triangle) and Fig 1b (cyan triangle) were taken are also shown. Fig. 1a and Fig. 1b were taken facing east. (b) The approximate location of the CARC in Moccasin, MT in Central Montana is marked by the red star. The State of Montana is also highlighted in red with a fill color of tan. (Basemap Image: © Google Tiles).**

## 3 Data and Methods

### 3.1 In Situ Measurements

The CARC was selected for NASA's SnowEx field campaign in the winter of 2020/2021 to study prairie snow as one of its main objectives. SnowEx efforts at the CARC included airborne L-band interferometric synthetic aperture radar (InSAR) flights from the Uninhabited Aerial Vehicle Synthetic Aperture Radar (UAVSAR) instrument, snow-on and snow-off UAV lidar observations, UAV

orthophotos and structure from motion (SfM), and ground-based snow observations including snow pits and snow depth transects (Palomaki and Sproles, 2023). For this analysis, we used UAV mounted lidar measurements of snow depth along with snow density measurements from snow pits to calculate spatially distributed SWE at the CARC.

Table 1 summarizes the snow depth properties and Fig. 3a shows the resulting digital snow models (DSM) from the 8 UAV lidar flights made in winter 2020/2021 across 8 different dates in our 1 km$^2$ study area (dashed black box, Fig. 2). The lidar data at the CARC were acquired by a contractor, DJ&A, P.C., using a 1,550 nm and a 905 nm wavelength laser (Woodley et al., 2024). The lidar measurements show how snow depth varies spatially and temporally within the CARC. The lidar flight

conducted on 15 January 2021 is considered our "no snow" baseline. Despite the large range in snow depth due to the snow drifts, the snow drifts typically covered less than 1% of the 1 km$^2$ area before February 2021. This includes a prominent linear north-south snow drift that formed adjacent to a windbreak in the western portion of the CARC. For this study, the digital snow model from the UAV lidar was divided into 2 m by 2 m pixels, for a total model domain of 500 pixels by 500 pixels. We

masked off any region with 0 cm snow depth as a "no snow" region. We note that root mean squared errors (RMSE) provided by the contractor were between 4 and 7 cm, possibly due to the winter stubble giving a false surface return (Palomaki and Sproles, 2023). We compared our DSM from 21 January 2021 to the pixel classifications made from an orthomosaic photo on the same day (Figs. 1d and 1e from Palomaki and Sproles, 2023), and the two show good agreement. However, our "no snow" masks

include some pixels that are classified as "Mixed" in Palomaki and Sproles, 2023, likely due to the shallow and discontinuous nature of the snow in these areas.

**Table 1: Snow depth (SD) and the snow covered area (SCA) statistics from the digital snow models from each of the 8 UAV lidar flights at the CARC. We report the average and maximum SD for each date. The SCA is reported as the percentage of the CARC**
**within the 1 km$^2$ research area is covered by snow and the percentage of the CARC covered by greater than 20 cm of snow.**

| Date | Avg. SD, Excluding Bare Ground (Avg. With Bare Ground) [cm] | Max. SD [cm] | SCA [%] | SCA, SD > 20 cm [%] |
|---|---|---|---|---|
| 15 Jan. 2021 | 5.3 (0.1) | 63.4 | 1.8 % | 0.2 % |
| 21 Jan. 2021 | 3.6 (1.6) | 96.7 | 45.1 % | 0.6 % |
| 22 Jan. 2021 | 3.8 (2.0) | 82.7 | 52.1 % | 0.5 % |
| 29 Jan. 2021 | 3.2 (0.9) | 82.8 | 28.1 % | 0.5 % |
| 17 Feb. 2021 | 8.8 (7.9) | 131.5 | 89.6 % | 5.0 % |
| 18 Feb. 2021 | 8.7 (7.6) | 131.0 | 87.1 % | 4.8 % |
| 24 Feb. 2021 | 5.5 (2.2) | 100.6 | 39.7 % | 2.4 % |
| 4 Mar. 2021 | 2.2 (1.3) | 80.4 | 60.1 % | 1.1 % |

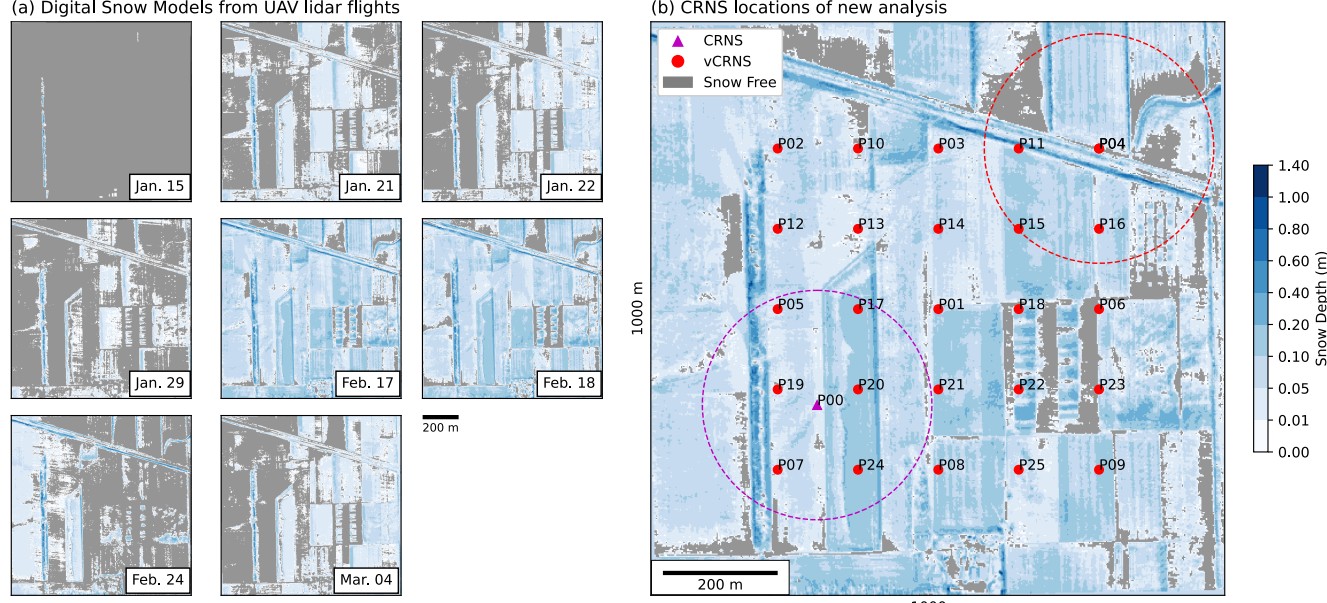

**Figure 3 (a)** Lidar digital snow maps (DSM) from the winter 2020-2021 NASA SnowEx Prairie Mission within the research domain (dashed black box in Fig. 2a). Gray regions indicate regions of no snow cover (SD = 0 cm). Color scale for snow is not linear. Smaller increments were included to show where extremely shallow snow is located at the CARC. **(b)** Map of locations of virtual CRNS points for URANOS simulations. The actual CRNS location is marked by the magenta triangle, with the calculated 171 m operational footprint (magenta dashed circle) of the CRNS from Woodley et al. (2024). The rest of the virtual CRNS (vCRNS) locations used in this analysis are marked by red circles, with one example virtual CRNS footprint shown in the red dashed circle in the upper right.

To calculate spatially distributed SWE from UAV snow depth, we used density measurements from snow pits measurements collected in the north-south snow drift in the western portion of the CARC research domain (Mason et al., 2024). Snow pits observations were collected on four dates: 20 January, 17 February, 24 February, and 5 March 2021. The snow pits revealed a bimodal snow density distribution, with a lighter snow layer (varying between approximately 100 kg m$^{-3}$ for newly fallen snow to slightly over 400 kg m$^{-3}$ late in the melt season) atop a denser basal layer (approximately 400-500 kg m$^{-3}$) Thus, we utilized a 2-layer density scheme to calculate spatially distributed SWE at the CARC, using snow density values derived from the snow pit measurements. The thickness of the lighter and basal snow layers on a given date was determined by differencing the lidar DSMs on different dates. These 2-layer snow density and depth maps were used to specify the "natural" snow cover conditions in the neutron transport simulations (section 3.2). The snow pit data are archived and freely available on the National Snow and Ice Data Center (NSIDC) Distributed Active Archive Center (DAAC). A more detailed summary of our methodology is provided in the Supporting Information from Woodley et al. (2024).

## 3.2 Ultra-Rapid Neutron Only Simulations

We analyzed the effects of the spatial heterogeneity of prairie snow on CRNS measurements through neutron transport modelling. Recently, CRNS studies have adopted the use of the Ultra Rapid Neutron-Only Simulation (URANOS), such as Brogi et al. (2022), Schattan et al. (2017), and Schrön et al. (2023). URANOS utilizes a Monte Carlo approach to simulate the neutrons and has been specifically developed for CRNS applications (Köhli et al., 2023). Millions of neutrons are generated from randomly distributed point sources within a user-defined area, and neutrons' path and interactions are tracked from its source to the point of detection through a ray-casting algorithm (Brogi et al., 2022; Köhli et al., 2023). URANOS can model 3-dimensional voxel-based geometries with defined materials by stacking multiple layers of either ASCII matrices or bitmap images to replicate important site characteristics (Köhli et al., 2023). For this analysis, we used URANOS v1.23, which is freely available for download at: https://gitlab.com/mkoehli/uranos/.

To examine how CRNS measurements change with the spatial distribution of snow, we ran 624 individual URANOS simulations: corresponding to each of the 26 virtual CRNS locations around the CARC (Fig. 3b), for each of the eight dates corresponding to the UAV lidar flights at the CARC, with three different snow distribution schemes on each date. The three different snow distribution schemes include two different sets of simulations using two different spatially uniform snow layers and a singular set of simulations using a "natural" or heterogeneous snowpack using DSMs derived from the UAV lidar and snow density (Fig. 3a). We also ran control simulations with completely snow-free conditions for each virtual CRNS locations. Our "natural" or heterogeneous model setups are similar to the simulations described in Woodley et al. (2024), with a stratified 2-layer snow density model as described in Sect. 3.1 and split into semi-regular layers (see colorbar on Fig. 3). However, our simulations also contain several important differences. First, we moved the virtual CRNS around our research domain to test how neutron counts would have been affected by the differing snow cover conditions around the CARC. A cylindrical virtual CRNS detector was placed at each of the 26 points on Fig. 3b and placed 2 m above the ground in URANOS. Each URANOS run simulated $10^8$ neutrons. The virtual CRNS was enlarged to a 9 m radius to improve detection statistics and supplied with a detector response function (provided in the URANOS GitLab repository) to simulate the sensitivity of the CRNS installed at the CARC, specifically a high-density polyethylene moderator of 25 mm thickness. To minimize the influence of soil heterogeneity and focus on the influence of snow variability, we chose to create a uniform 30 cm soil layer with the average of all soil measurements. In the field, soil samples for soil moisture and bulk density were collected at 5 cm depth intervals up to a total depth of 30 cm and at six cardinal directions at three different radii (approximately 25 m, 75 m, and 200 m) from the CRNS instrument (Woodley et al., 2024). Because this analysis moves the simulated CRNS instrument around the CARC where other soil moisture measurements were not made, we chose to average the soil measurements for our uniform soil layer. As in Woodley et al. (2024), soil moisture, atmospheric pressure, and other important parameters listed in Table 2 were kept constant to allow direct comparisons of model simulations due to changes in snow distribution and to remove the need to correct counts based on differing hydrogen pools.

**Table 2: Atmospheric and soil parameters used in our URANOS simulations. These values were unchanged from each set of heterogeneous and uniform snow runs.**

| Parameter | Value |
|---|---|
| Number of Neutrons [-] | 100000000 |
| Air Humidity [g m$^{-3}$] | 3.341 |
| Atmosphere Depth [g cm$^{-3}$] | 888.809 |
| Soil Moisture (first 30 cm) [%] | 21% |
| Soil Bulk Density (first 30 cm) [g m$^{-3}$] | 1.087 |
| Soil Porosity (first 30 cm) [%] | 56% |

For the uniform simulations, a chosen volume of snow water was evenly distributed in the research area, creating a uniform snow layer. We created two uniform snow layer schemes based off: a)
230 the average amount of snow water in the 171 m operational footprint around the CRNS detector and b) the average amount of snow water across the entire 1 km$^2$ study domain. The 171 m operational footprint of the CRNS is a site-specific value calculated at the CARC using "no snow" URANOS simulations from Woodley et al. (2024). While we used a constant value for the CRNS footprint in this study, the actual operational footprint of a CRNS is dependent on the amount of moisture present in the
235 environment. We derived the uniform snowpack thickness by dividing the total amount of snow water volume by the snow density of hard coded material values of different snow types in URANOS. Depending on the amount of snow water per pixel, we chose to model the snowpack using the built-in material codes for snow: 240, 241, and 242, which has density values of 0.03 g cm$^{-3}$, 0.1 g cm$^{-3}$, 0.3 g cm$^{-3}$, respectively, to create a snow layer with uniform thickness and density (see MaterialCodes.txt in
GitLab repository, link in Sect. 3.2).

From the different URANOS simulations, we also calculated SWE from the modeled neutron counts. We followed our methods from Woodley et al. (2024) to calculate modeled SWE from URANOS. SWE calculations were made using Eq. (1) (Desilets, 2017) using our modeled neutron counts from URANOS simulations

$$SWE = -\Lambda \ln \frac{N - N_{wat}}{N_\theta - N_{wat}}. \tag{1}$$

$N_\theta$ is the calibration neutron count, from the "snow-off" reference date of 15 January 2021. $N$ is the neutron counts corresponding to the dates of the subsequent seven "snow-on" lidar flights at the CARC (21 Jan. 2021 to 4 Mar. 2021). The attenuation length ($\Lambda$) was calculated to be 4.8 cm from previous literature (Desilets et al., 2010). $N_{wat}$ is the counting rate over an infinite depth of water and can be
calculated using Eq. (2):

$$N_{wat} = 0.24 N_0, \tag{2}$$

where 0.24 is an assigned constant value (Desilets, 2017; Desilets et al., 2010). $N_0$ is the theoretical counting rate over dry soils:

$$N_0 = \frac{N_\theta}{\frac{a_0}{\theta_g \rho_{bd} + a_2} + a_1}, \tag{3}$$

where $a_0 = 0.0808$, $a_1 = 0.372$, and $a_2 = 0.115$ (Desilets et al., 2010; Desilets, 2017). Usually, $N_\theta$ in Eq. (3) is multiplied by a correction factor, $F(t)$, to correct for solar activity, atmospheric pressure, and humidity. However, as all our model simulations used the exact same meteorologic conditions, our correction factor was set to 1. $\theta_g$ is the sum of gravimetric soil water content, soil mineral lattice water and water equivalent of soil organic carbon, and $\rho_{bd}$ is the soil bulk density, which were obtained from in situ soil samples.

## 3.3 Comparisons with Gridded SWE Products

To evaluate whether CRNS SWE has potential value for future remote sensing missions or gridded datasets, we compared our CRNS SWE and UAV lidar SWE to several gridded SWE products, which are available at several spatial resolutions. We chose the Western United States UCLA Daily Snow Reanalysis (hereafter UCLA-re, ~500 m resolution, Fang et al., 2022), the Snow Data Assimilation System (SNODAS, 1 km resolution, National Operational Hydrologic Remote Sensing Center, 2004) from National Oceanic and Atmospheric Administration's National Weather Service National Operational Hydrologic Remote Sensing Center, and the Daily 4 km Gridded SWE (hereafter UA, 4 km resolution, Broxton et al., 2019) from the University of Arizona.

The UCLA-re dataset is generated from assimilation data with Landsat fractional snow cover area and other input data such as meteorological forcings from the Modern-Era Retrospective analysis for Research and Applications, version 2 (MERRA-2) (Margulis et al., 2019). A Bayesian analysis is performed on prior estimates of snow states and fluxes using a land surface model and snow depletion curves (Margulis et al., 2019). SNODAS provides daily gridded estimates of SWE for the conterminous United States using a snow model, which is forced by downscaled numerical weather predictions (Clow et al., 2012). Digitally available airborne, satellite, and ground-based snow data are then assimilated into the model to provide a best estimate of near real-time snow estimates (Clow et al., 2012; Driscoll et al., 2017). The UA dataset provides SWE and snow depth estimates by assimilating snow station data such as the snow telemetry (SNOTEL) network and precipitation and temperature data using the gridded PRISM model (Zeng et al., 2018). For each gridded dataset, we chose the pixel that included the CARC. Only the SWE for the UCLA-re data was aggregated and averaged within a 2-pixel by 2-pixel region, to obtain an area that is similar to the 1 km$^2$ area of the CARC. All gridded datasets are freely available for download at the National Snow and Ice Data Center (last accessed: 3 October 2024).

## 4 Results and Discussion

## 4.1 Neutron Modelling

Figure 4 shows the differences between the URANOS simulations with a heterogeneous snowpack and 171 m average uniform snowpack for neutron counts (Fig. 4a) and SWE (Fig. 4b) for all 8 lidar flight dates and 26 virtual CRNS locations on Fig. 3b. Neutron counts are on average 1.8% higher in the heterogeneous runs compared to the uniform runs with a root mean squared difference (RMSD) of 2.6 %. When we calculated the SWE using these URANOS runs and Eq. (1), SWE would be

underpredicted in the heterogeneous runs with a mean bias percent error (MBPE) = -19.9 % and a RMSD = 35.3 %. We found similar trends comparing URANOS simulations with a heterogeneous snowpack and the CARC average uniform snowpack (not shown). Neutron counts were 1.9% higher in the heterogeneous runs and an RMSD of 3.1 %. SWE were biased towards the uniform runs with an MBPE of -23.2 % and a RMSD = 42.7 %. For both comparisons, we colored each data point in Fig. 4 by the percentage of bare ground (i.e., the ratio of the area of no snow cover (SD = 0 cm) to the total area of the 171 m radius footprint of each virtual CRNS detector). Generally, we found neutron counts were similar between the heterogeneous and uniform runs (both 171 m and CARC average SWE) at higher percentages of bare ground within the operational footprint of the CRNS. The opposite trend was true for SWE.

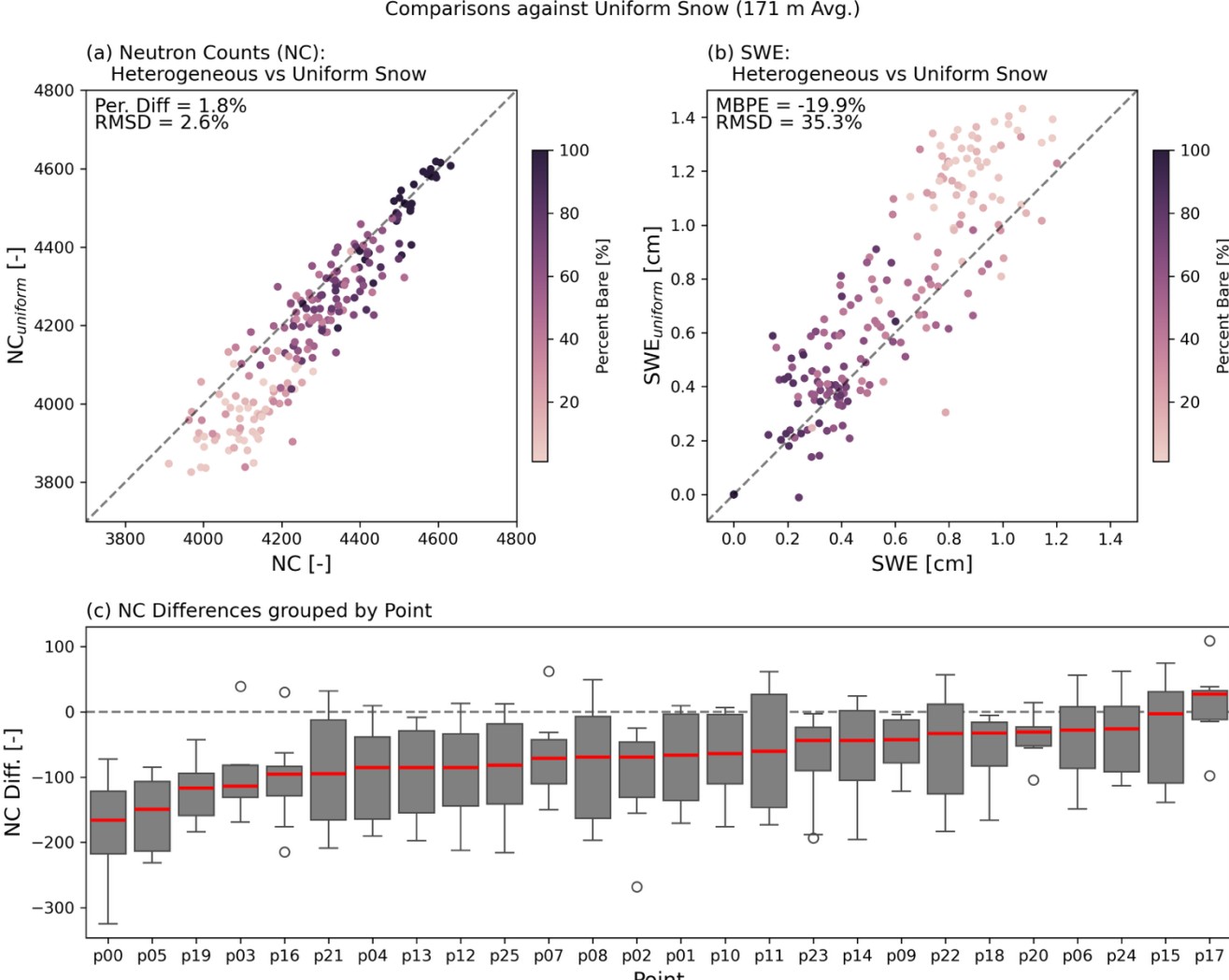

Figure 4 Scatterplot comparing (a) neutron counts and (b) SWE for the heterogenous snow runs (x-axis) against uniform snow runs (y-axis) that use the average SWE of the 171 m radius footprint surrounding the virtual detector across the 26 virtual CRNS


locations for the 8 lidar flight dates. Points are colored by the percentage of bare ground by area within the 171 m footprint of a CRNS. (c) Boxplots showing the difference between the heterogenous and uniform snow runs for each virtual CRNS location (shown in Fig. 3b), where each box contains the eight URANOS simulations corresponding to the eight UAV lidar flights at the CARC.

We grouped the differences in neutron counts between the heterogenous and uniform snow model runs (with CRNS footprint average SWE) across all dates by virtual CRNS location, to determine which locations had the largest and smallest differences in neutron counts (Fig. 4c). The largest differences were found in points P00, P05, P19, and P03. Points P00, P05, and P19 are the 3 closest

locations to the large snow drift in the western portion of the study area. P03 (top row, center in Fig. 3b) is also located near snow drifts that formed due to topographical changes near train tracks that cross the CARC. The lowest errors were found in points P17, P15, P24 and P06. The commonality between points P17, P15, P24, and P06 were likely relatively uniform snow cover surrounding the virtual CRNS for most of the dates. P17 and P24 were in the same field directly to the left of P00, which had relatively

uniform snow trapped from the field around most of the dates during winter 2020/2021. P00, P05, P19, and P03 had much more variable snow cover surrounding the virtual CRNS, with the large snow drift on one side and bare ground on the other for most dates in winter 2020/2021.

Comparing the heterogeneous runs to the uniform runs with CARC average SWE allows us to evaluate which virtual CRNS locations were most reflective of the CARC average. The locations with

the smallest neutron count differences were points P20, P07, P06, and P19. The locations with the largest neutron count differences were points P13, P23, P14, and P10. Interestingly, points P20, P07, and P19 are the three points clustered around the actual CRNS instrument at the CARC. P06 was not located near the original CRNS but had some snow cover through most of January and February. P13, P14, and P10 were also similarly clustered close together (NW quadrant) closer to the train track snow

drifts. We theorize that these points sampled too many snow drifts or too little snow throughout the winter.

One might assume that neutron counts between the uniform and heterogeneous simulations should be comparable because both have the same total snow water volume within the operation footprint of the CRNS. However, it appears that the distribution of the snow water and bare ground

patches among fallow fields, crop stubble, and shelter belts around the CARC has a considerable effect on CRNS, as shown in Schattan et al. (2019). Figure 4c suggests that snow drifts closer to the CRNS affect neutron counts the most, leading to the largest differences in neutron counts compared to a uniform snow scenario. We found that differences in neutron counts between the uniform and heterogeneous runs (hereafter $\Delta NC_f = NC_{uniform} - NC_{heterogeneous}$) were positively correlated with the

percentage of bare ground within the operation footprint of the CRNS in the heterogenous scenario (i.e., spatially varying snow distribution derived from the UAV lidar and snow density), with statistical significance ($r = 0.454$, $p < 0.05$). This correlation partly arises from the fact that we are comparing similar model runs when the bare ground percentage is close to 100%, leading to minimal differences in neutron counts. Differences in $\Delta NC_f$ between the uniform and heterogeneous snowpacks increases with

more snow covering the ground, and enhanced variability of snow depths within the CRNS footprint. To verify, we computed additional correlation metrics between the $\Delta NC_f$ and snow depth variability within a CRNS footprint – namely the standard deviation and the range (difference between max. and min. snow depth). We found statistically significant negative correlations between $\Delta NC_f$ and snow

depth standard deviation (r = -0.70, p < 0.05) and $\Delta NC_f$ and snow depth range (r = -0.60, p < 0.05). The
negative correlations are due to $\Delta NC_f$ being mostly negative since $\Delta NC_{heterogeneous} > \Delta NC_{uniform}$. These
results similarly suggest that higher amounts of snow lead to increased heterogeneity (e.g., snow drifts
and bare ground patches) which creates the high $\Delta NC_f$.

To test whether snow drifts do in fact play a large role in neutron count differences, we focused
on model comparisons for 15 January 2021, to isolate the effects of large snow drifts on CRNS
measurements. Figure 5 shows the differences between heterogeneous runs (i.e., spatially varying snow
distribution derived from the UAV lidar and snow density) and the uniform runs (i.e., uniform snow
distribution) from 15 January. On this date, most of the CARC was snow-free except for some isolated
patches of extremely shallow snow and the large snowdrift in the western portion of the study domain
(top left panel of Fig. 3a, and Fig. 5b). Most virtual CRNS locations resulted in neutron counts from the
heterogenous and uniform runs that were within 1 % of error from each other. However, points P00,
P05, P07 and P19 yielded large differences of greater than 100 neutrons (approximately 3 % error).
These four points are also the closest to the snow drift on 15 January 2021 (see Fig. 5b).

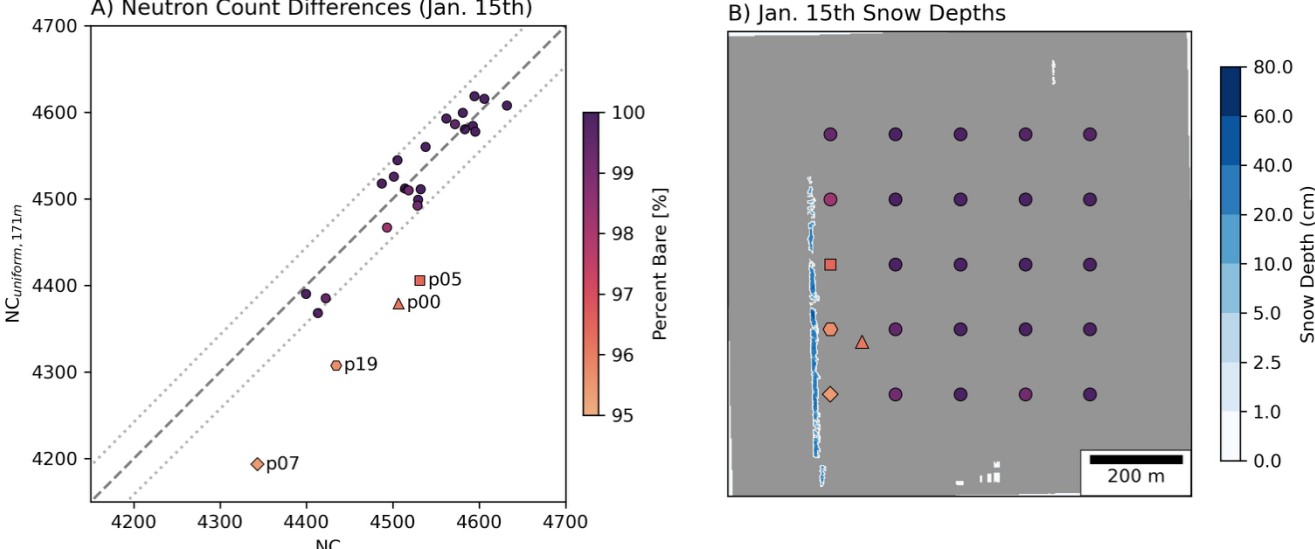

**Figure 5 (a) A scatterplot comparing neutron counts from the uniform runs (y-axis) against the heterogeneous runs (x-axis) for 15
January 2021, the near-no-snow baseline, with the exception being the large north-south snow drift in the western portion of the
study area (same as Fig. 4a). The points are colored by the percent of bare ground within the 171 m footprint of the CRNS but
using a different scale. While most points fell near the one-to-one line (black dashed line) and within a 1% error, four virtual
CRNS locations yielded large differences in neutron counts: P00 (triangle marker), P05 (square marker), P07 (diamond marker),
and P19 (hexagonal marker). (b) Map of the snow depth from the 15 January 2021 UAV lidar flight, shown in the colorbar. The
snow drift is the slim blue linear feature on the left (western) portion of the study area. The virtual CRNS locations in URANOS
are shown in circles, while the actual CRNS location from winter 2020-2021 is shown in a triangle (as in Fig. 3b). The four points
with the largest neutron count differences are marked in magenta.**

Figure 6 compares how the neutron counts change with relation to the snowpack variability at
P00, P05, P07 and P19. We calculated the percent change between the heterogeneous and uniform runs
(171 m average) where the neutron model domain was divided into twelve sectors of equal angle from

the virtual CRNS detector. We noticed skews in neutron origins due to the relation of the model geometry, namely the position of the virtual detector and the source geometry. Virtual detectors placed closer to the edges of our domain had neutron origins that were skewed towards the center of the
domain. Therefore, we limited the neutron counts to within a 200 m radius of the virtual detector only for the results shown in Fig. 6(a)-6(d). The radial plots in Fig. 6 shows the percent change in neutron counts from the uniform runs to the heterogeneous runs in each sector on 15 January. P07 (Fig. 6a) saw the biggest percent change between the no-snow (right of N-S line) and snow side (left of N-S line) with an average percent change of 5 % in neutron counts compared to 1.6% change, respectively. We
observed a similar but smaller trend in P05 (Fig. 6c) with an average 3.2% change on the no-snow side and 2.3% change on the snow side. In both P19 (Fig. 6d) and P00 (Fig. 6b), we observed larger changes on the snow side compared to the no-snow side. P00 had a 5.3 % change on the snow side compared to a 2.4 % change on the no-snow side. P19 had a 3.9 % change on the snow side and a 2.1 % change on the no-snow side. The different trend in P00 neutron counts are likely explained by the longer distance
away from the snow drift (Fig. 6f) leading to extreme difference in the snowpack around the CRNS. Many studies have shown that CRNS is extremely sensitive to its immediate surroundings (Köhli et al., 2015; Schrön et al., 2017). In the case of P00, it seems that the latter has a lesser influence on neutron counts compared to points P05, P07, and P19, which were much closer to the snow drift. These results highlight that CRNS neutron counts are the result of the interaction between the spatial sensitivity of the
CRNS and the spatial snow distribution. The differences between P05, P07, and P19 are likely caused by the breaks in the snow drift as it first formed. P07 (Fig. 6e) was placed next to a longer, contiguous section of the snow drift compared to P05 (Fig. 6g), which reduced the neutron counts on the snow side for P07. We observed similar breaks in P19. Overall, all of our model results are likely influenced by the extremely shallow nature of the snowpack at the CARC, leading to differences in neutron counts
that are less than 10% of the detected neutrons, making this correlation analysis difficult to discern.

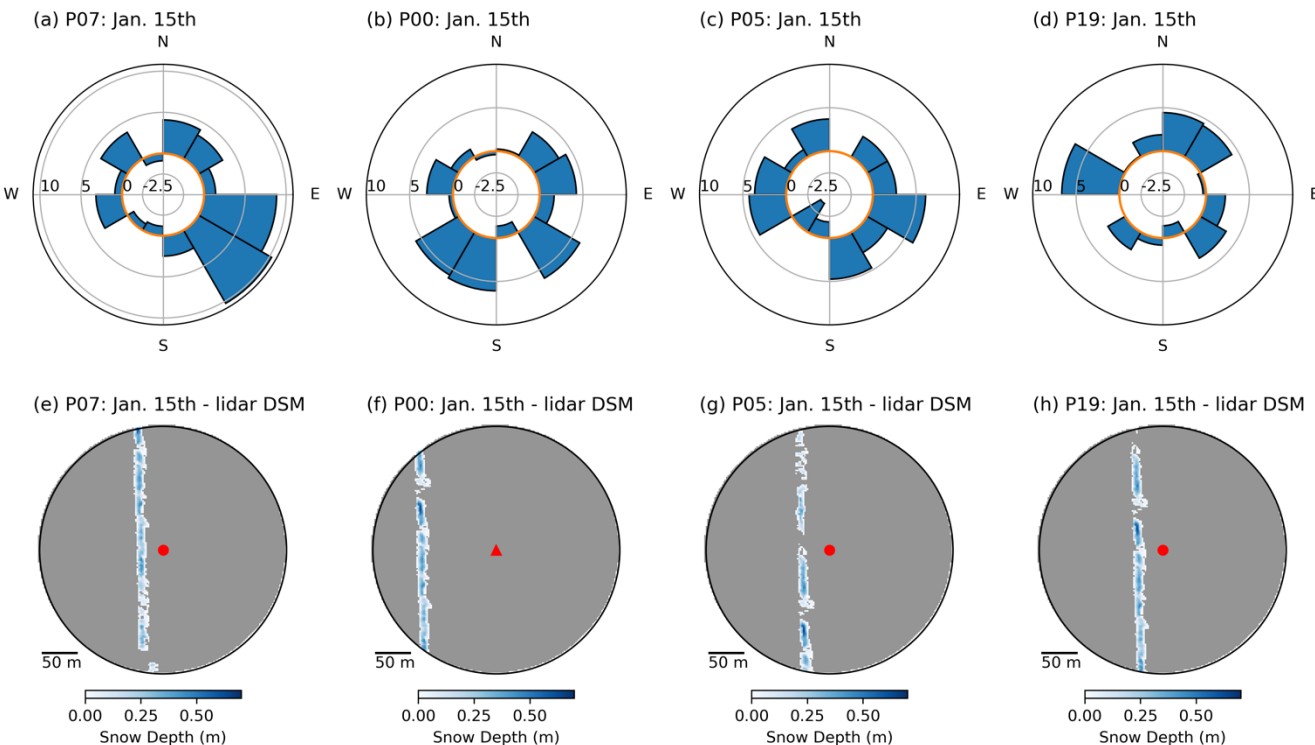

**Figure 6** Percent changes in neutron counts of the heterogenous runs from the uniform runs for 12 sectors around the virtual CRNS location for the 4 points identified in Fig. 3: (a) P07, (b) P00, (c) P05, and (d) P19. The orange line on panels (a)-(d) marks no change in neutrons counts in the heterogeneous runs from the uniform runs. The snow distribution on 15 January 2021 is shown for each point on panels (e)-(h) to contextualize the differences.

## 4.2 CRNS Spatial Representativeness

To supplement these findings, we conducted a secondary analysis to evaluate the spatial representativeness of CRNS SWE at our prairie site compared to the observations that might have been collected from a more traditional snow scale SWE instrument. In most cases, CRNS or other SWE instruments would be deployed in hopes of capturing the average snow conditions representative of a large area. In order to do this, we averaged the lidar-derived SWE DSMs for each of the eight UAV flights to 1 $m^2$ spatial resolution. We calculated the kernel density of all of these 1 $m^2$ SWE pixels to understand the full distribution of SWE across the study site, where each pixel represents a possible SWE measurement that could have been collected by a naively located snow scale or snow pillow (of measurement area equal to 1 $m^2$). Then, we applied the CRNS spatial weighting function from Woodley et al. (2024) to each of these pixel locations (actually, every 4th pixel to increase computational efficiency), using a wraparound boundary to remove edge effects from pixels close to the boundary of the study site. This allowed us to retrieve a distribution of synthetic CRNS SWE estimates across the entire CARC.

We acknowledge that this analysis is naive in that it assumes that the CRNS spatial weighting function would be constant across the entire study site. In reality, the spatial sensitivity of CRNS can

change with snow spatial distribution and magnitude, and soil moisture distribution and magnitude, among other factors. The wraparound boundary also means that none of the CRNS SWE estimates from
this analysis, especially those near the boundaries of the study area, are truly reflective of the "true" SWE that would be observed by CRNS at the same location within the site.

However, it does mean that each CRNS SWE estimate is derived from the same lidar-derived SWE data, which reflects a spatial snow distribution representative of a prairie site. Lastly, this analysis assumes that a snow scale or snow pillow would exactly measure the SWE in each given location.
However, this is unlikely to be true given that snow will likely accumulate differently on a smooth artificial surface versus the natural ground surface, especially in the windy, shallow snow conditions typical of the prairie. In summary, this analysis is not as rigorous in reproducing CRNS behavior as the URANOS simulations presented above. Still, it does provide a first-order estimate of the spatial representativeness of CRNS SWE estimates at a prairie site versus more conventional, smaller-footprint
SWE instruments.

Figure 7a shows the kernel density distribution of synthetic SWE estimates from the CRNS locations across the entire CARC (blue), compared to the distribution of "Snow Scale" 1 m$^2$ lidar-derived SWE pixels from the entire CARC (red) for an example date of 29 January 2021. This date was more than one week after the most recent snow event, allowing for wind redistribution, sublimation, and
potentially melting of the snow during the intervening period. The spatial mean lidar-derived SWE for the entire CARC is shown in the vertical, black dashed line. A similar plot is shown in Fig. 7b for 17 February 2021, soon after a large snow event (and the most pronounced snowpack of the season). In

both cases, the CRNS SWE distribution is shifted closer to the CARC average, compared to the 1 m² "Snow Scale" SWE distribution.


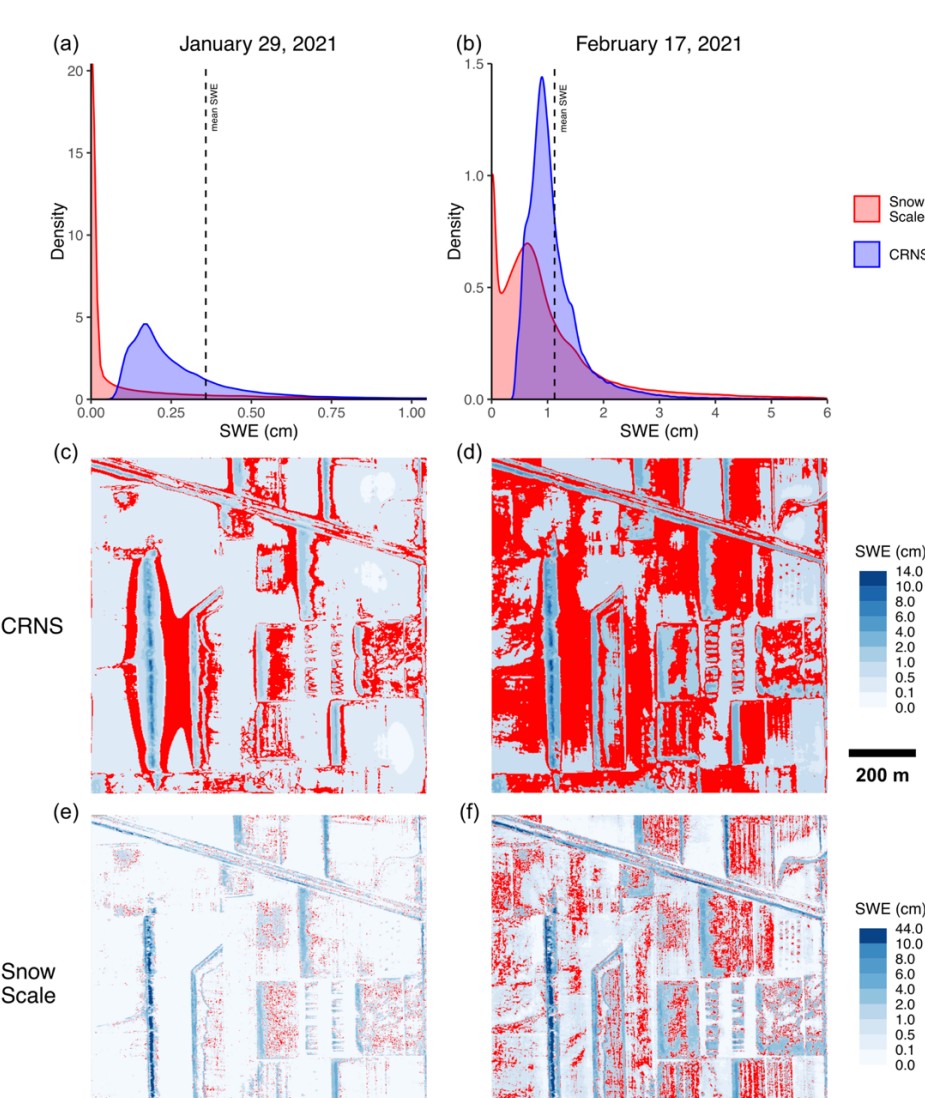

Figure 7 Simulation of the spatial representativeness of aboveground CRNS at the CARC versus a snow scale or pillow of area 1m². (a) and (b) Probability density functions of the SWE observed by synthetic CRNS (blue) versus a synthetic snow scale or pillow of pixel size 1m by 1m (red) for 29 January and 17 February 2021, respectively. The vertical dashed line shows the mean SWE of the entire study 1 km² area. It is evident on both dates that the probability density of CRNS SWE estimates is shifted closer to the areal mean. (c) and (e) show the areas where the CRNS and 1m "Snow Scale" are within +/- 25 % of the mean SWE of the entire study area (red pixels), respectively, for 29 January 2021. The underlying blue color map shows the SWE estimate from the given synthetic SWE measurement method, as calculated from the lidar-derived SWE DSM. This results in different color scale limits for the CRNS (c) than for the synthetic snow scale (e) because the CRNS measures SWE over a larger spatial footprint, which effectively smooths out the SWE distribution. (d) and (f) show the same information for 17 February 2021. Generally, the CRNS is representative of a larger proportion of the study area and the representative areas are more contiguous, compared to the 1m resolution synthetic snow scale or pillow.



For 29 January, the CARC average SWE was 0.4 cm. 23 % of the CRNS locations were within +/- 25 % of the CARC average, while only 5% of the 1 $m^2$ pixels were within that same range. For February 17, the CARC average SWE was 1.1 cm, and 50 % of the CRNS locations and 20 % of the 1$m^2$ pixels were within +/- 25 % of the CARC average, respectively. Across all dates (excluding January 15, 2021, which had very spatially limited snow cover), this analysis indicated that the percentage of the CARC study area for which a CRNS would return a SWE estimate within +/- 25 % of the CARC average ranged from 21-50 %, while the 1 $m^2$ pixels ranged from 5-20 % of the CARC. In summary, our first-order analysis indicated that a naively sited CRNS was 2.3 to 5 times more likely to return a SWE estimate within +/- 25 % of the large-scale spatial average than a similarly sited SWE sensor with a footprint of 1 $m^2$.

These results are shown spatially in Figs. 7 c&e, where Fig. 7c shows the map of synthetic CRNS SWE estimates, and Fig. 7e shows the lidar-derived SWE at 1 $m^2$ resolution for the example date of 29 January 2021. In both maps, locations that returned a SWE value within +/- 25 % of the CARC average are shown in red. The representative areas for CRNS are more extensive and spatially contiguous, while the representative 1 $m^2$ "Snow Scale" pixels are fewer and less spatially contiguous. The same maps are shown for 17 February 2021 in Figs. 7 d&f. In this case, a larger proportion of the CARC is representative of the large-scale CARC average in both maps, and the CRNS similarly shows more extensive and more contiguous representative areas. These results indicate that CRNS provides value for large-scale SWE estimates in the prairies, and well suited to measure SWE in prairie environments compared to the conventional, smaller-footprint sensors. It appears that the optimal locations to site CRNS in prairie snow distributions like the CARC are in locations of low snow accumulation near areas of high snow accumulations (e.g. snow drifts). This makes sense, as most of the CARC area exhibits low snow accumulation, while only a small portion experiences higher snow accumulation, and CRNS are most sensitive to the area immediately surrounding the instrument. Through a combination of design and happenstance, our actual CRNS at the CARC (point P00 on Fig. 3) is located within a representative region for all lidar dates (with the exception of 15 January 2021, which had very spatially limited snow cover).

**4.3 Comparison against Gridded SWE Estimates**

To show the value of accurate CRNS measurements to future remote sensing missions, we compared our CRNS SWE estimates and currently available gridded snow products to the areal mean lidar-derived SWE and snow depth for the entire 1 $km^2$ study area. Figure 8 shows comparisons of SWE (Fig. 8a) and snow depth (Fig. 8b) products at similar magnitudes of scale (see Sect. 3.3 for details). We also plotted our CRNS SWE time series at the CARC from Woodley et al., (2024) (see Fig. 5a in Woodley et al., 2024). In January and March 2021, all gridded SWE products had no SWE. This contrasts with the average CARC SWE from the UAV lidar DSMs (red squares on Fig. 8a) and URANOS simulations (grey boxplots on Fig. 8a), and our CRNS SWE times series (green line, Fig. 8a), which all indicate that snow is present. In February, the UCLA-re and SNODAS predicted more peak SWE on the 17 and 18 February 2021 compared to our average CARC SWE, with SNODAS almost double our CARC SWE

estimates. The UA SWE produced quite similar estimates to our CARC SWE in February, before underpredicting SWE starting in March.

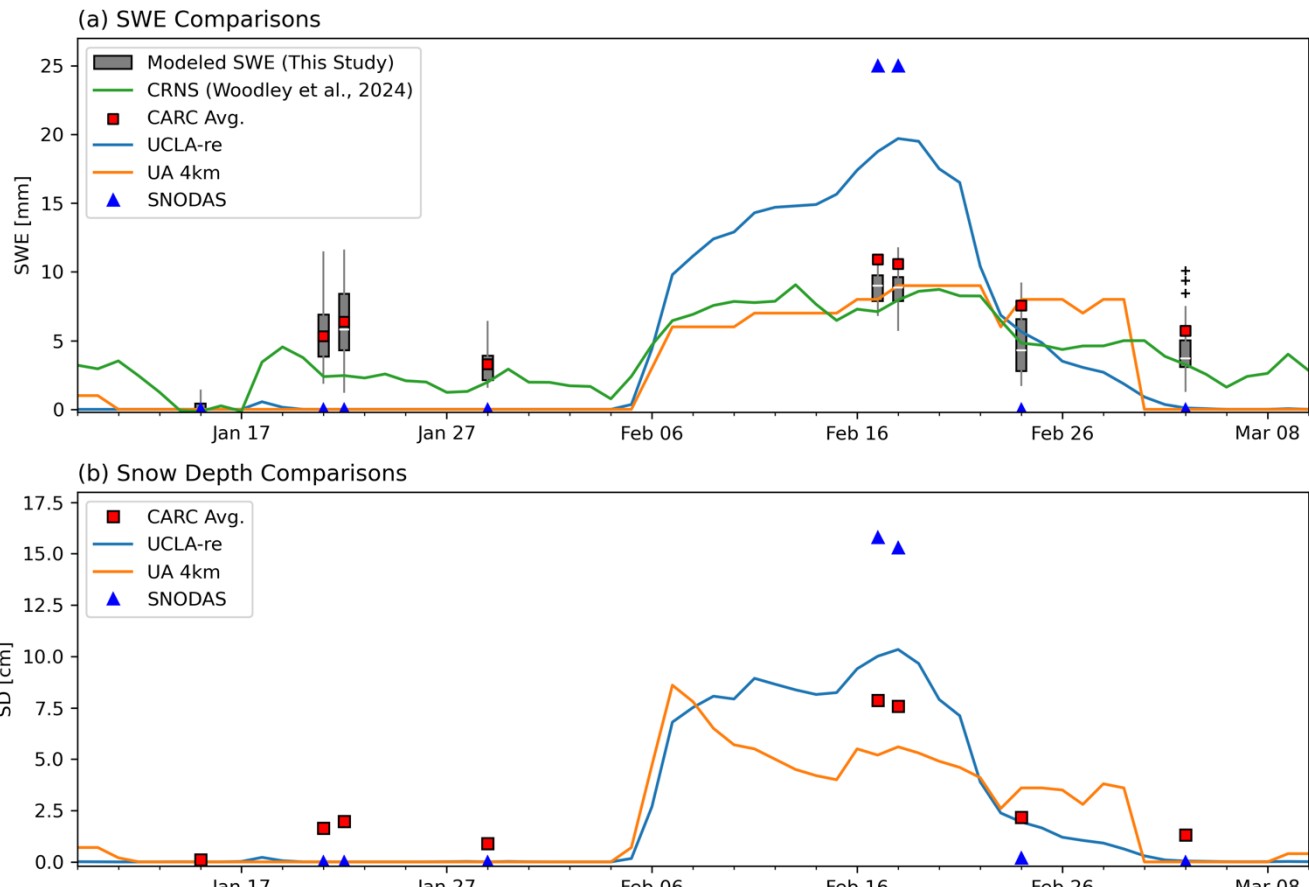


**Figure 8 Comparisons of (a) SWE estimates and (b) snow depth estimates from gridded products and in situ measurements between 9 January to 10 March 2021. In (a), time series of the UCLA Snow Reanalysis (blue line), UA SWE (orange line), daily mean CRNS SWE (green line) from Woodley et al., (2024; blue line from Fig. 5a) are shown. Daily SNODAS SWE estimates for each of the dates corresponding to a lidar flight are shown as blue triangles, and an averaged CARC SWE for each digital snow**


**model (DSM) for the 1 km² study region are plotted as red squares. URANOS modeled SWE estimates from this study for each date are plotted as grey boxplots to illustrate the variability of SWE within our study region. Snow depth from the same sources are shown in (b), except for CRNS and URANOS, which do not estimate snow depth.**

The differences in SWE products are likely due to aggregation with different resolution and meteorological forcings. Sub-grid variability is shown to be very important in estimating the SWE in a

prairie environment, where the average SWE can be either grossly under or overpredicted. Past studies have indicated that SNODAS is unsuccessful at capturing the snow spatial variability in regions with persistent winds like the prairies (Lv et al., 2019). Our results indicate that similar issues can occur with snow depth. Figure 8b plots a similar graph, except showing the changes in snow depths for January to March 2021. Snow depth shows a similar pattern, where all gridded products lack snow in January 2021

and March 2021, and snow depths are detected for February 2021. SNODAS overestimates the snow

depths compared to our average CARC snow depths on 17 and 18 February 2021, while underestimating snow depths for all other dates, despite having similar spatial resolution (1 km for SNODAS and a 1 km aggregate for lidar CARC SWE).

While Fig. 8a shows that SWE estimates from the UA 4km data are more reliable in February 2021, Fig. 8b shows that the accuracy of both the UA and UCLA-re snow depth estimates vary depending on the winter months. UA 4km underestimates snow depth for mid-February 2021, while the UCLA-re overestimates snow depth. However, by the end of February 2021, this relationship is flipped with the UCLA-re predicting similar snow depth to our lidar DSM average and the UA 4km overpredicting snow depth. The timing of snow accumulation from all three models also does not seem

to line up with some of our in-situ measurements. UCLA-re shows a brief accumulation event between the 15 January 2021 UAV flight and the 21 January 2021 UAV flight, and coincident with a known snowfall event between 18-19 January 2021 (see Supporting Information for Woodley et al., 2024). However, snow disappears quickly after the snowfall event. Lower estimates of mean SWE and SD are expected for larger spatial resolutions due to increased aggregation (Blöschl, 1999).

Our analysis shows that CRNS has utility for improving SWE estimates in prairie environments, and other environments with shallow, heterogeneous snowpacks. CRNS measurements have already shown this utility in mountain regions. Integration of CRNS SWE into models, alongside remote sensing data, has reduced error spread in the Austrian Alps (Schattan et al., 2020). CRNS has the potential to increase the coverage of SWE monitoring sites, where currently used technologies within

snow monitoring networks like SNOTEL may not be optimal, such as the northern Great Plains. Previous research has shown that large errors in SWE were due to subpixel SWE variability of the Northern Great Plains (Tuttle et al., 2018). However, we hope that future planned satellite missions such as NISAR, armed with similar instrumentation used in the CARC during SnowEx 2021 (Palomaki and Sproles, 2023) can improve efforts to monitor snow in this relatively under-instrumented region.

**4.4 Assumptions and Limitations of this Study**

For this analysis, we made several key assumptions and simplifications from actual field conditions during winter 2020/2021. One key simplification concerned soil moisture. As mentioned in Sect. 3.2, we kept soil moisture spatially uniform and constant across all our model simulations due to a variety of logistical complications. In situ soil moisture measurements were collected at the CARC after the winter

season in May 2021, due to delivery of the CRNS instrument after first snowfall (Woodley et al., 2024). These soil measurements were also taken at a maximum of 200 m away from our CRNS instrument, while our URANOS simulations cover the entire 1 km$^2$ area. While soil moisture was continuously monitored at nine locations throughout the winter of 2020/2021 using soil moisture probes, these data were not informative because the soil temperature dropped below 4 ºC (at which point water's dielectric

properties change) for the top 0.5 m of soil for nearly the entire winter (Woodley et al., 2024). The heterogeneity of the underlying soil moisture will have a great effect on CRNS measurements and neutron counts, possibly even overcoming the contribution of the snowpack due to the shallow nature of the snowpacks in the prairie. Snowmelt events throughout the winter could also impact CRNS measurements throughout the winter, which may also impact soil moisture depending on the coupled

frozen ground dynamics. Our aim was to show how CRNS measurements were affected by snowpack spatial distribution alone and what considerations need to be taken before siting a CRNS to obtain SWE.

Another important assumption was our initial conditions, namely our $N_\theta$, the calibration neutron count (see Eq. 1 and 3), which we took from 15 January 2021. Typically, a CRNS is calibrated by choosing a $N_\theta$ value before the start of the winter season, when SWE = 0 (Desilets, 2017). Again, due to logistical constraints mentioned previously, we were not able to obtain a baseline neutron count during snow-free conditions. Between the time period when the CRNS was installed at the CARC on 22 November 2021 and when we conducted our "snow-off" lidar flight on 15 January 2021, the CARC was never completely snow-free (Woodley et al., 2024). Our $N_\theta$ value from 15 January may be lower than a calibration value chosen before the start of the winter season due to the proximity of the prominent north-south snow drift. A lower $N_\theta$ would affect the SWE values that we have calculated in this study and our CRNS time series (green line in Fig. 8). However, with less than 2 % of the CARC covered in snow on 15 January 2021 and only 0.2 % of it covered in deep snow (see Table 1), we do not expect the choice of $N_\theta$ to be a large source of bias in our CRNS SWE estimates. Modeled SWE calculated using a completely snow-free baseline (grey boxes Fig 8a) and the January 15th baseline (Fig. 4b) differed on average by 0.05 cm.

## 5 Conclusions

A neutron transport modelling study at an agricultural site in the Northern Great Plains of Montana has shown that the spatial variability of shallow and heterogeneous snowpack affects CRNS measurements. Our URANOS simulations with heterogeneous snowpack tended to have increased neutron counts compared to simulations with a uniform snowpack with similar snow water volume. We partly attribute these increases in neutron counts to bare ground patches around the CRNS with the heterogeneous snowpack, similar to previous studies such as Schattan et al., (2019). However, we acknowledge that the spatial sensitivity of the sensor may play a large role in these differences as well, since our virtual CRNS locations were placed in areas of lower snow accumulation. How snow is distributed should be considered when siting aboveground CRNS instruments in areas of high snow spatial heterogeneity, even for very shallow snowpack like that at the CARC, if the goal is for the instrument to be representative of the large-scale spatial average. In prairie sites characterized by wind scoured fields and spatially limited snow drifts, CRNS instruments should be placed in areas of low snow accumulation that are nearby higher snow accumulation areas. However, a naively sited CRNS instrument (i.e., with no knowledge of the snow distribution) is still 2 to 5 times more likely to be representative of the large-scale average SWE than a more conventional, smaller footprint SWE sensor such as a snow scale or snow pillow. Comparisons with gridded SWE products show that CRNS has the potential to improve SWE estimates in prairie snow, when compared to lidar-derived SWE from the site. Our study focuses solely on the effect of snow distribution on CRNS, but spatial variability of soil moisture is also important to consider, especially in shallow snowpack areas such as the prairie, where the effect of soil moisture distribution on CRNS measurements may be of comparable magnitude to that of snow

distribution. This highlights the need for further research in semi-arid prairie environments like the Northern Great Plains, where water use efficiency and snow capture are of great agricultural interest, and more rigorous studies of CRNS applications in shallow, heterogeneous snowpacks.

## Code and Data Availability

Code and data used in this analysis will be made available through GitHub at https://github.com/heyjoekim/carc_crns_spatial and archived on Zenodo at https://doi.org/10.5281/zenodo.15530868. Snow pit data from the CARC are available to download from the NSIDC DAAC (https://doi.org/10.5067/QIANJYJGRWOV).

## Author Contribution

HK did the formal analysis and wrote the initial draft of the research conceptualized by ST and HK. ES installed the CRNS and completed the drone flights at the CARC. ST and ES reviewed and edited the paper.

## Competing interests

The contact author has declared that none of the authors has any competing interests.

## Acknowledgements

We would like to acknowledge the National Aeronautics and Space Administration (NASA) Terrestrial Hydrology Program for their support of the SnowEx field campaign, which is a primary source of data for this analysis. We would also like to thank Dr. Eric Sproles' graduate and undergraduate students for their participation for their participation in the ground-based snow sampling during winter 2020-2021. We would also like to thank Dr. Markus Köhli and Dr. Cosimo Brogi for their help in understanding how to operate and set up the URANOS simulations used in this study.

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
