# Peer review of "Influence of Snow Spatial Variability on Cosmic Ray Neutron SWE: Case Study in a Northern Prairie"

_EGUsphere, 2025_

## Referee Comment (RC3)

**Review on "Influence of Snow Spatial Variability on Cosmic Ray Neutron SWE" by Kim et al.**

*The study of Kim et al. on the "Influence of Snow Spatial Variability on Cosmic Ray Neutron SWE" evaluates in a comprehensive analysis the effect of snow distribution on the signal of a cosmic ray neutron sensor at an agricultural prairie site in Montana. The authors apply a threefold analysis to find general recommendations on the position of a CRNS probe for area-representative measurements. In a first approach the effect of snow cover is tested in Neutron Simulations under homogeneous and heterogeneous snow conditions. A second step compares the area-representativeness of CRNS to virtual snow scale/pillow observations with a smaller footprint. Finally, the CRNS-derived SWE signal is compared to daily satellite-based SWE products. The limited amount of studies on above-snow CRNS and the need for area-average SWE observations for remote sensing product calibration and integration make this study very valuable.*

**General comments**

The outline of the manuscript is well structured and easy to follow. By testing the effect of snow cover on different virtual CRNS sensor locations, the authors find that the in-field CRNS sensor is located at an area-representative location, before evaluating satellite-based SWE observations against the CRNS-derived SWE measurements. By calibrating the SWE conversion with a snow free reference value (15 January), the pre-snow moisture level of the site is considered. However, two major points require reconsideration: 1) The effect of partial snow cover has been noted in line 89-90, and addressed in line 314 to 332 (with SCA ranging from 1.8 % to 89.6 %). Partial snow cover does not seem to affect the NC signal in this study. Since this finding is contrary to previous studies, the results of the correlation analysis should be shown and addressed in more detail. The lacking indication of snow-free areas in the snow cover maps provides the impression of a closed snow cover (see comments on figure 3 ) and misses that the footprint of many virtual sensors is only partially snow covered or snow-free. 2) Line 260 to 263 note boundary effects in the URANOS model, but section 3.2 does not specify the boundary conditions that were chosen. For clarity, the study should provide further information and the distance between the outer virtual sensors and the boundaries of the URANOS domain.

**Minor comments**

- The title should indicate that the analysis covers a case-study in a prairie environment.
- The major outcome that is outlined in the abstract from line 17 to 20 should not be indicated as a logical consequence. It rather seems that study 4.1 shows that CRNS is influenced by snow drifts and study 4.2 shows that an area average can be obtained by placing a sensor in the proximity of a snow drift. However, figure 8 c) and d) shows that an area average may also be obtained in a location afar from snow drifts, meaning that both findings are true, but don´t condition each other.
- Line 186-188: It is acceptable to use a constant footprint size, but the footprint dependency on the amount of present moisture (i.e. snow) should be briefly discussed.
- Analysis 4.1 distinguishes between uniform snow thickness scenarios, computed from the SWE average of the CRNS footprint and the SWE average of the study domain. In the sub-studies, outlined from line 240 to 283, it becomes not clear, which of the two scenarios have been used.
- The results and discussions around Figure 4 and 5 seem straight forward. However, it is questionable if the "snow-free" day is a good choice for an analysis of the effect of snow cover. If the SWE average is based on the CRNS footprint in this analysis, almost all virtual detector locations are compared under completely snow-free conditions, except for the sensors close to the remaining snow patch ("snow drift"). Choosing a day with a more prominent snow cover (e.g. 17 February) would be more relevant.
- Results and discussions around Figure 6 and 7 would benefit from additional information on how much each virtual detector was affected by fractional snow cover throughout the study. This would strengthen the discussion, which seems to evaluate the complexity of snow cover within the footprint area from visual inspection.
- The analysis of section 4.2 and 4.3 give a great added value to the study. While results of 4.2 are partially mentioned in the abstract (l. 20-22) and a hint on 4.3 is provided in the introduction (l. 94-95) they appear hidden and should be more clearly visible, in both abstract and introduction.
- The analysis in 4.1 shows that CRNS measurements on the "snow-off" day (January 15) were affected by the snow drift, presumably lowering the $N_\theta$ that was chosen for the SWE conversion. The effect on the converted SWE signal should be briefly discussed in 4.3.
- Consider rephrasing line 480 to 482 for better logical reasoning and more clarity.

**Illustration remarks**

- Figure 1:
    - For clarity, the position and viewing direction of these images could be marked in Figure 2.
- Figure 3:
    - A different color should be applied to snow-free areas to allow for a differentiation into areas of heterogeneous snow cover and areas of partial snow cover.
    - The choice of an exponential color scale is reasonable, but should be better indicated in the legend (e.g. color bar with exponential color distribution, instead of even increments)
    - The images miss a scale bar. A dashed line that indicates the domain outline as in Figure 2 would be additionally interesting, as well as the distance of the outer virtual detector locations to the domain boundary.
- Figure 5:
    - For consistency, the color scale in e) to f) should be the same as in the previous figures (white indicating low snow and blue indicating high snow accumulation). Further, the SD maps miss a scale bar.
    - Since the findings at P00 and P19 are contrary (larger changes on the snow side) to the findings at P07 and P05 (larger changes on the no-snow side) besides the similarity in snow distribution, P19 should also be presented in this figure.
- Figure 6 & 7:
    - The figure should indicate which scenarios were included in the analysis (all except 15 January).
    - Coloring the scatter plot after the snow cover fraction within the corresponding virtual detector footprint may add valuable insights.
- Figure 8:
    - The choice of red as a color for agreement seems counter intuitive. Green may be a better choice (the significance of that color needs to be indicated in the legend).
    - All maps miss a scale bar.
    - The exponential character of the SWE color bar should be displayed with exponential color increments.

---

## Author Comment (AC2)

We thank Nora Krebs and Dr. Paul Schattan for their comments on this manuscript. Our response to each comment is highlighted in blue.

Minor comments

- The title should indicate that the analysis covers a case-study in a prairie environment.
- We agree our title is very broad and should be more definitive. Along with the previous comment from Prof. Köhli, we will change the title to highlight that this is a case study in a prairie environment. Our proposed revised title is "Influence of Snow Spatial Variability on Cosmic Ray Neutron SWE in Northern Prairies".

- The major outcome that is outlined in the abstract from line 17 to 20 should not be indicated as a logical consequence. It rather seems that study 4.1 shows that CRNS is influenced by snow drifts and study 4.2 shows that an area average can be obtained by placing a sensor in the proximity of a snow drift. However, figure 8 c) and d) shows that an area average may also be obtained in a location afar from snow drifts, meaning that both findings are true, but don´t condition each other.
- We agree that lines 17 to 20 is not an exact logical consequence from our results in this study. We will revise these lines to better reflect the results of our analysis.

- Line 186-188: It is acceptable to use a constant footprint size, but the footprint dependency on the amount of present moisture (i.e. snow) should be briefly discussed.
- Thank you for this comment. We agree that the amount of moisture around a CRNS alters the effective footprint. We will add a brief discussion to the manuscript about the importance of footprint size and its dependency on moisture.

- Analysis 4.1 distinguishes between uniform snow thickness scenarios, computed from the SWE average of the CRNS footprint and the SWE average of the study domain. In the substudies, outlined from line 240 to 283, it becomes not clear, which of the two scenarios have been used.
- We apologize that parts of the manuscript were not clear about which SWE scenario we used. For lines 240 to 276, the analysis around Figures 4 and 5 used the model results from the heterogeneous (i.e., "natural") snow distribution and the SWE average from the given CRNS footprint. We will revise these sections to increase clarity.

- The results and discussions around Figure 4 and 5 seem straight forward. However, it is questionable if the "snow-free" day is a good choice for an analysis of the effect of snow cover. If the SWE average is based on the CRNS footprint in this analysis, almost all virtual detector locations are compared under completely snow-free conditions, except for the sensors close to the remaining snow patch ("snow drift"). Choosing a day with a more prominent snow cover (e.g. 17 February) would be more relevant.

- Results and discussions around Figure 6 and 7 would benefit from additional information on how much each virtual detector was affected by fractional snow cover throughout the study. This would strengthen the discussion, which seems to evaluate the complexity of snow cover within the footprint area from visual inspection.

- We are responding jointly to the two comments above, since they seem related. Our intent in Figures 4 and 5 was to illustrate the influence of spatially limited, high SWE snow drifts on our CRNS results. We felt that January 15[th] was ideal for this because of its lack of snow cover outside of the snow drift.  The snow distribution from other dates would include this effect, but it would be overprinted by the influence of snowpack elsewhere in the CRNS footprint.  We accept the criticism that this example doesn't necessarily show all of the considerations that influence the CRNS model results.

  We also accept the feedback on Figures 6 and 7. We agree that the analysis can further benefit from how the virtual detector was affected by fractional snow cover throughout the study. We will add qualitative comparisons to our discussion. To that end, we also propose changing the order of the results presented in Section 4.1. We will present the complete results first (current Figures 6 and 7), which are all affected by the heterogeneous snow distribution, and add a color scale to the points to reflect the fractional snow cover. Then, we will discuss how the snow drift also affects our results (current Figures 4 and 5).

- The analysis of section 4.2 and 4.3 give a great added value to the study. While results of 4.2 are partially mentioned in the abstract (l. 20-22) and a hint on 4.3 is provided in the introduction (l. 94-95) they appear hidden and should be more clearly visible, in both abstract and introduction.

- Thank you for this comment. We agree sections 4.2 and 4.3 are important to this study and should be highlighted in our abstract and introduction. We will edit our abstract and introduction to include these results.

- The analysis in 4.1 shows that CRNS measurements on the "snow-off" day (January 15) were affected by the snow drift, presumably lowering the $N\theta$ that was chosen for the SWE conversion. The effect on the converted SWE signal should be briefly discussed in 4.3.
- We agree and we will include a brief discussion of the converted SWE signal into 4.3. We will also revise Fig. 9 to include the SWE calculated from the bare ground conditions.

- Consider rephrasing line 480 to 482 for better logical reasoning and more clarity.
- Thank you for pointing out this lack of clarity. We will rephrase lines 480 to 482.

Illustration remarks

- Figure 1:
    - For clarity, the position and viewing direction of these images could be marked in Figure 2.
    - Thank you for this comment. We will add markers to Figure 2 that will clarify the position and viewing direction of our images on Figure 1.

- Figure 3:
    - A different color should be applied to snow-free areas to allow for a differentiation into areas of heterogeneous snow cover and areas of partial snow cover.
    - We agree that marking the snow-free areas and areas of partial snow cover may be beneficial and clearer to readers. We will change Figure 3 to include these no-snow masks (see example figures below, with no snow areas shown in gray). However, we must note that the snow was very shallow for many of our observation dates, and orthophotos were only available for one of the dates, so we cannot be completely certain about the fractional snow cover percentage across all dates. The uncertainty that exists with our lidar measurements were outlined in Woodley et al. (2024), with RMSE values between 4 and 7 cm. The high RMSE values were likely from the wheat stubble giving a false return. There is potential that an incorrect threshold snow depth for delineating snow-covered vs. snow free areas could drastically change the fraction of snow cover within the study area. However, we compared our masks (using 0 cm snow depth as "no snow") with a snow cover class analysis of the CARC conducted by Palomaki and Sproles (2023). We are including Figure 1d and 1e into this discussion from Palomaki and Sproles (2023) which

shows that creating a snow cover mask using a threshold snow depth of 0 cm matches the snow cover class analysis from an orthomosaic photo on 21 Jan. We will include discussions of this uncertainty in our manuscript as well.

[Figure]

*Figure R1. Lidar snow depths (SD) in m for 21 January 2021. Snow free pixels are shown as grey. Snow free pixels are any pixels with a SD equal to 0 m.*

[Figure]

**Snow Mask = 4.0 cm, 21 Jan. 2021**

1000 m

*Figure R2. Lidar snow depths (SD) in m for 21 January 2021. Snow free pixels are shown as grey. Snow free pixels are any pixels with a SD less than 4 cm (0.04 m). This threshold was chosen due to the uncertainty in the lidar flights.*

[Figure]

*Figure R3. Figure 1(d) from Palomaki and Sproles (2023). An orthomosaic image of the CARC on 21 January 2021 with a spatial resolution of 10 m.*

[Figure]

*Figure R4. Figure 1(e) from Palomaki and Sproles (2023). The snow cover classes at the CARC at a spatial resolution of approximately 5 m.*

- o The choice of an exponential color scale is reasonable, but should be better indicated in the legend (e.g. color bar with exponential color distribution, instead of even increments)
- o We understand the reviewers point that showing the colorbar on an exponential scale would be a clear signal to the reader that the colorbar is not linear. However, we found that the exponential scale makes the tick mark values harder to read, as it is harder to differentiate the colors when the ticks are compressed into the upper portion of the colorbar. While the scale is nonlinear, we think that showing a set number of categories makes the snow depth more interpretable to the reader. We have attached an example of the figures below. We found that the differences between the colormaps are very minor. However, we note that it is not possible to show a value of 0 with a log distribution, so we do lose any values between 0 and 1 cm (0.01 m). For these reasons, we have retained the current color scale on our figures. However, we will make sure to note the irregular color scale in the figure captions so that readers are aware.

[Figure]

*Figure R5. Lidar DSM of snow depths at the CARC for 21 January 2021. The colorbar is the same colorbar as the manuscript.*

[Figure]

*Figure R6. Lidar DSM of snow depths at the CARC for 21 January 2021. The colorbar distribution is now exponential.*

- o The images miss a scale bar. A dashed line that indicates the domain outline as in Figure 2 would be additionally interesting, as well as the distance of the outer virtual detector locations to the domain boundary.
- o We thank the reviewers for their comment. We will add a scale bar to the maps. To address the second part of the comment, all of these maps are within the dashed domain outline in Figure 2, which is why we did not plot it in Figure 3. The footprint for p04 was added to illustrate what the comment suggested. We understand that this was not clear. We will add a label to the x-axis on Fig. 3b like the ones in Figures R5 and R6 to show it was a 1000m and will clarify in the caption of Figure 3 that our study area was 1000 m by 1000 m.

- Figure 5:
  - o For consistency, the color scale in e) to f) should be the same as in the previous figures (white indicating low snow and blue indicating high snow accumulation). Further, the SD maps miss a scale bar.
  - o In the original Figure 5, we reversed the colormap because the SD maps would blend into white background of the figure but kept the colors consistent. We will alter the figure so that the colors are consistent. Also, we will add scale bars to our SD maps.

  - o Since the findings at P00 and P19 are contrary (larger changes on the snow side) to the findings at P07 and P05 (larger changes on the no-snow side) besides the similarity in snow distribution, P19 should also be presented in this figure.
  - o We originally left off P10 from figure 5 because we felt the individual panels would have been too small to make out any details. We will include P19 in Figure 5 to highlight the contrary findings.

- Figure 6 & 7:
  - o The figure should indicate which scenarios were included in the analysis (all except 15 January).
  - o We apologize for the lack of clarity. All scenarios were used in this figure. We will clarify this in the text, caption, or figure.

  - o Coloring the scatter plot after the snow cover fraction within the corresponding virtual detector footprint may add valuable insights.

- We agree that coloring the scatter plot by the snow cover fraction may yield valuable insights. We will revise the figure accordingly.

- Figure 8:
    - The choice of red as a color for agreement seems counter intuitive. Green may be a better choice (the significance of that color needs to be indicated in the legend).
    - We understand the reviewer's comment about our choice of color, however, selecting a decent color to contrast with the blue is a challenge. Green may not be the best choice if we want to have a figure that is colorblind friendly. We can potentially change it to orange, but feel the red color serves the purpose of being visible and distinguishable from the background colormap.

    - All maps miss a scale bar.
    - We will add a scale bar to all figures that show maps.

    - The exponential character of the SWE color bar should be displayed with exponential color increments.
    - This comment is similar to a previous comment about exponential color increments. Please refer to our response above that includes Figures R5 and R6.

References:

Palomaki, R. T. and Sproles, E. A.: Assessment of L-band InSAR snow estimation techniques over a shallow, heterogeneous prairie snowpack, Remote Sensing of Environment, 296, 113744, https://doi.org/10.1016/j.rse.2023.113744, 2023.

---

## Author Response (AR1)

Response to Reviewer 1:

We thank Dr. Markus Köhli for the prompt, detailed feedback on our manuscript. We are providing a response to each comment in blue. We have altered our comments from the public comments during the discussion to reflect the actual changes made to the manuscript when appropriate. No other edits were made to the responses.

Quality:

- The manuscript is written in a straight forward way and can be easily followed. Some sentences might appear to be rather long and complex. The figure quality is good. The references are well organized.

- One shortcoming of this study is that soil moisture was not consistently measured throughout the reference period. Sect. 3.2, citing Woodley et al., 2024, implies that in-situ samples were collected only once. With only considering snow heterogeneity the whole approach of the analysis of different influences on the CRNS signal has a synthetic character. The authors do not hide that fact, but they also do not clearly discuss it.

It would be completely out of scope to integrate soil moisture distributions here as well, yet, the findings from the authors may by such either be increased or smeared out. Soil moisture is most often correlated with the patches of snow, which means if areas with shallow snow pack additionally are more wet, the heterogeneity increases. On the other hand, all relative (SWE) signal variations depend on the underlying soil moisture. This can either decrease or increase the effect of snow. Such should clearly be discussed in order to provide a guide for non-CRNS experts for how to interpret the results.

We agree that in these cases, especially with shallow, heterogeneous snowpacks, soil moisture will impact the CRNS signal. We were able to collect in-situ soil samples once at the end of the winter of 2020/21, but these data provided the spatial distribution of soil moisture only at a single moment in time. In addition, soil moisture sensors were installed in nine locations throughout the study area during winter 2020/21, with sensors at 3-4 depths between 0-50 cm in each location. However, these data were not informative because the ground temperature dropped below 4 degrees C (at which point water's dielectric properties change) for the top 0.5 m of soil for nearly the entire winter. We highlighted this drawback in Woodley et al. (2024); however, we acknowledge that this should have been made clear in this manuscript as well.  In summary, we do not feel that we have sufficient information about the spatiotemporal distribution of soil moisture during the study period to include it in our simulations. We chose to average our in situ soil moisture measurements and used this average as a constant soil moisture for our

research domain in all simulations. We have added section 4.4 to discuss this and other important assumptions and limitations of our study.

- how is SWE calculated from the LIDAR measurements? Do the authors assume constant snow density?

Snow density measurements from manual snow pits were applied to the lidar snow depth measurements in order to calculate SWE. These snow pits were conducted in the north-south snow drift in the western portion of the study area. Based on the snow pit observations, we adopted a 2-layer density scheme (a lighter snow layer atop a denser, basal snow layer) where the density of each layer was determined from the snow pits and the variation in depth of each layer across the study domain was determined by differencing lidar snow depth maps on different dates.

A detailed process was outlined in Supplementary Information for Woodley et al., 2024, which this manuscript also uses. We will highlight how SWE was calculated by adding a summary of these methods to the revised manuscript and link the appropriate methods and data sources to this manuscript. We have added a paragraph in Section 3.1 that summarizes how SWE was calculated.

- Is simplified weighting function (B1) from the appendix of DOI 10.5194/hess-21-5009-2017 not working for the SWE distribution (it does not have, to, the question is in fact more related to the Woodley publication).

For Woodley et al. (2024), we found that the simplified weighting function (B1) provided a fit that was adequate but not satisfactory for the distribution of neutrons from our URANOS simulations. However, we were not entirely satisfied with the match to our data; specifically B1 slightly underpredicted neutron counts at short radii (20-60m) relative to our URANOS model output. As a result, we opted to fit our own weighting function to the model output, based on the B1 formulation, and found that our method provided the best fit out of several weighting function types that we tested.

- The comparison analysis is somehow confusing in the way it is structured. Like for soil moisture, you can assume, that there is a horizontal weighting function for SWE. There is no such of a systematic analysis existing in literature, the authors presented one solution in Woodley et al., 2024. The authors compare uniformly assumed SWE with the actual SWE distribution patters (and Eq.(1)). They compare equal weighting with simulations of detectors at 25 virtual locations. The authors find in their analysis that inhomogeneities in the SWE distribution lead to deviations from the assumed equal weighting. That part is discussed quantitatively in detail. They then switch from count

rate considerations to spatial representativeness, where they compare a lidar SWE evaluation with an analytically weighted SWE. Here they find that the CRNS weighted SWE is more close to the area average.

Why are those results or evaluations not combined into one discussion? First, the neutron simulation also directly provides the neutron density over the study site. This density can be transformed to a SWE value can easily be compared to other ones. The virtual detectors you only need to describe where neutrons come from, to trace for example the origins individually. For just the count rate the entire detector layer in URANOS provides the values at each pixel in the domain. Secondly, after the probably a little bit too detailed quantitative discussion you repeat the comparison of SWE average vs. heterogeneous by only changing the averaging radius from 171 m to the whole domain. Whereas that is a part to be considered it takes a lot of room, especially as the results are not so different. Thirdly, there is switch from counts and comparison to averages to which points in their domain correspond to the average and therefore are more representative by now including another method which is analytical weighting.

In summary, that part should be condensed and restructured.

We agree that some of our results can be condensed and restructured for greater clarity. We have attempted to do this in the revised version of the manuscript.

Regarding your proposed alternate approach to this analysis, we somewhat understand what you are suggesting, but we are not completely certain about all aspects. Your suggestion seems like a good strategy for understanding the spatial distribution of neutron density across the study area, which can then be used to evaluate the spatial representativeness of individual pixels compared to the areal average. Our approach in this analysis was different, but find it equally effective.

The goal of the analysis is to directly compare how a CRNS instrument will behave in the conditions of a synthetic uniform and natural snow distribution, which we did. In our analysis, we also wanted to understand how representative a CRNS estimate is of a large region, to inform comparison to remote sensing or gridded data, or provide information to applied CRNS users that can be used to understand the effective representativeness of CRNS of a large area and for siting CRNS instruments. We felt that this approach required the use of virtual CRNS detectors, given that we were uncertain if a direct analogy could be made between counts from the opaque virtual CRNS detector and a transmissive detector layer in URANOS. However, we feel that both of these approaches are viable ways to address the goals of the analysis.

Specific comments:

Comments to the figures:
- Labels A and B in the figures are referred to in the caption as (a) and (b).

Thank you, Prof. Köhli, for catching these mistakes in our figures. We have edited both the figures and the captions to be consistent throughout the manuscript.

- Caption Fig. 6: "surrounding the virtual" -> "surrounding the virtual detector".

Thank you for catching the typo in Fig. 6. We have edited the caption in the revised manuscript. (now Fig. 4)

- Fig. 7 seems to not add a lot more to what can already be seen in Fig. 6

We somewhat agree that Fig. 7 does not differ significantly from Fig. 6. While we believe that the point of the analysis in Fig. 7 still has some value in demonstrating the representativeness of the CRNS of large areas, we see that the similar results do not add enough to justify another figure. Thus, we have removed Fig. 7 from the manuscript.

- Fig. 8: in a) and b) the dashed line could receive a label in order make more clear what you want to compare. The label "1 m" in the legend is not very descriptive.

We agree that the dashed lines and the labels for Fig. 8 could be confusing. We have added a "mean SWE" label to the dashed lines Fig. 8 (now Fig. 7a and Fig. 7b) and changed the legend label from "1 m" to "Snow Scale" to hopefully better communicate the analogy represented in the simulation.

- Fig. 9: convert the axis to normal dates. DOWY is used nowhere else in the manuscript.

We have edited the x-axis to normal dates in Fig. 9 (now Fig. 8) in the revised manuscript.

- Fig. 9: The "CRNS timeseries" does not match anything which is plotted in the reference Woodley et al., 2024.

The "CRNS timeseries" corresponds to the daily averaged SWE values from the actual CRNS counts shown in Fig. 5a of Woodley et al. (2024). The axes in our Fig. 9 differ slightly from the Woodley et al. (2024) figure. The y-axis of Fig. 9 is in mm instead of cm, and our x-axis in Fig. 9 includes a shorter time span.

- Fig. 9: The authors include the "CRNS timeseries" - do they mean SWE derived from CRNS?

We apologize for any ambiguity. For Fig. 9, we wanted to provide context to what the CRNS instrument installed at the CARC measured and compare them to the gridded products. The "CRNS timeseries" corresponds to the daily averaged SWE values from the actual CRNS counts measured by a CRNS instrument at our research site from Woodley et al. (2024). We have edited the figure and description to include more clarity regarding what the CRNS time series is.

Title:
- the reviewer thinks that the title "Influence of Snow Spatial Variability on Cosmic Ray Neutron SWE" overstates the results as the title is too general, although the research topic presented is a very broad basis for the SWE analysis. It should include that it primarily conceptualizes SWE analysis as derived from their site.

We agree that our title is overly broad and general. We changed our title to the following: Influence of Snow Spatial Variability on Cosmic Ray Neutron SWE: Case Study in a Northern Prairie.

Comments to the text body:
Introduction
- The overview is very well organized and includes the relevant literature. The different parts are, however, somehow disconnected. It begins with agricultural land-use changes, moves into dryland cropping techniques and then shifts to snow without clear linkage.

Thank you, Prof. Köhli, for this comment on the text. We have edited the introduction and attempt to make better connections between the several aspects of the introduction in the revised manuscript.

- l72: Hydrogen 'trap' (the technical term would be 'absorb') free neutrons if neutrons are thermalized. The CRNS signal attenuation is mainly due to slowing down of neutrons by hydrogen, i.e. losing energy through elastic collisions.

Thank you for correcting the language on our CRNS explanation. We have corrected our revised manuscript to better explain the CRNS signal attenuation.

Data and Methods

- The description of the NASA SnowEx field campaign at CARC is informative, but the connection between different measurement techniques (InSAR, UAV lidar, SfM, snow pits, etc.) is not well worked out given the specificities of each. Considering as the comparison of CRNS SWE with various gridded datasets (UCLA-re, SNODAS, and UA) is in general an important validation step and also undertaken in this study.

Thank you for the comments. We wanted to explain what was conducted during the NASA SnowEx campaign, but didn't make clear connections on how this field campaign intersected with this analysis. We will clarify this section and try to highlight the various gridded datasets in our revised manuscript. We have included which data our analysis uses and features.

Results and Discussion

- l198: "N_theta is the calibration neutron count, from the "snow-off" reference date of 15 January 2021." That is of course not a real calibration, it is an approximation for the calibration, which is associated with significant systematic uncertainties. One would need to for example look at the entire CRNS timeseries in order to try to understand (or justify) this choice. Even if you think, that this is a 'good enough' choice this needs to be discussed very clearly, as later on in the text you make very detailed numerical comparisons of which some directly relate to the choice of this value.

We agree that our choice of N_theta is an approximation of a calibration. We made this decision because it was not possible for us to collect calibration soil samples prior to winter snow accumulation due to delayed delivery of the CRNS instrument. We included this discussion in Woodley et al., 2024, but we will add discussion in this manuscript as well. This discussion was also added to the newly added section 4.4.

- l263: How significant are the relative differences mentioned here and shown in Fig. 5? The reviewer assumes, for example based on Fig. 6, that the number of neutrons counted rather low given the small relative differences. So that analysis might not be precise enough for sub-percent statements (i.e. 3.16%)?

You are correct that the relative differences are rather low, and the degree of specificity in the percentiles is a bit overstated. We will round the percentages to the nearest percent in the revised manuscript.

- l288: As stated above, the overestimation might simply be a result of not using any weighting function, but there might other effects playing a role, too.

Respectfully, Prof. Köhli, we are somewhat uncertain what you mean in the comment.

Specifically, we are unsure which weighting function you are referring to. The results discussed in this section were modeled outputs, where we directly compared model runs with a uniform versus heterogeneous snow layer. No spatial weighting function was applied to the model outputs. With everything else being equal in the model simulations, these "overestimations" are likely from the differences in the snow layers. We think that the "overestimation" may result from the fact that a large portion of the study area had low or zero snow accumulation, making it more likely that the virtual CRNS detectors would experience higher count rates in the heterogeneous snow simulation due to a large percentage of the neutrons originating close to the instrument. Additionally, in this manuscript (and Woodley et al., 2024) we have not applied a spatial weighting function to the model outputs

If we misinterpreted or did not adequately respond to your comment, we would welcome clarification.

- l353: you mention 624 simulations but not how you arrive at that number, given that, the reader is left wondering what that means.

Thank you, Prof. Köhli, for pointing out this omission. We will include a breakdown in the methods of how we arrived at the 624 number (26 points x 3 snow layer scenarios x 8 dates) in the revised manuscript.

Conclusion:
- l487: "a naively sited CRNS instrument (i.e., with no knowledge of the snow distribution) is still 2 to 5 times more likely to be representative of the large-scale average SWE than a more conventional (...)" - that statement is not found in the discussion before.

Respectfully, this statement was included in the end of Section 4.3 (l408). We will make these results more visible in the text, since this is a major takeaway from this analysis that we want to highlight.

Technical comments:
Typography:
- Equations are part of the text body. Therefore, they follow the interpunctuation. That means a dot after Eq. (1), not before, and a comma after (2) and (3).

- Tables: Try using the same number of decimals and align numbers either left or, preferably, right.

Thank you, Prof. Köhli, for the technical comments. We will make the appropriate edits to the punctuation around the equations and the table.

References:
- This manuscript refers to for the (previously already published) data. However, the DOI which is provided in Woodley et al., 2024 for the CRNS SWE data (https://doi.org/10.5067/NJR0AMMOHZ4E) does not work.

Thank you for pointing out the DOI is broken. This dataset is still in the process of publication with NSIDC since the Woodley et al., 2024 paper. The data will be available at this link once the process is complete.

Meanwhile, the raw CRNS data was also uploaded to the GitHub at: https://github.com/heyjoekim/carc_crns and archived on Zenodo at https://doi.org/10.5281/zenodo.11648961. The code to correct and calibrate our data is also provided in the same links.

Response to Reviewer 2:

We thank Nora Krebs and Dr. Paul Schattan for their comments on this manuscript. Our response to each comment is highlighted in blue.

Minor comments

- The title should indicate that the analysis covers a case-study in a prairie environment.
- We agree our title is very broad and should be more definitive. Along with the previous comment from Prof. Köhli, we will change the title to highlight that this is a case study in a prairie environment. Our revised title is "Influence of Snow Spatial Variability on Cosmic Ray Neutron SWE: Case Study in a Northern Prairie".

- The major outcome that is outlined in the abstract from line 17 to 20 should not be indicated as a logical consequence. It rather seems that study 4.1 shows that CRNS is influenced by snow drifts and study 4.2 shows that an area average can be obtained by placing a sensor in the proximity of a snow drift. However, figure 8 c) and d) shows that an area average may also be obtained in a location afar from snow drifts, meaning that both findings are true, but don´t condition each other.

- We agree that lines 17 to 20 is not an exact logical consequence from our results in this study. We have revised these lines to better reflect the results of our analysis.

- Line 186-188: It is acceptable to use a constant footprint size, but the footprint dependency on the amount of present moisture (i.e. snow) should be briefly discussed.
- Thank you for this comment. We agree that the amount of moisture around a CRNS alters the effective footprint. We will add a brief discussion to the manuscript about the importance of footprint size and its dependency on moisture. We have added a line in section 3.2.

- Analysis 4.1 distinguishes between uniform snow thickness scenarios, computed from the SWE average of the CRNS footprint and the SWE average of the study domain. In the substudies, outlined from line 240 to 283, it becomes not clear, which of the two scenarios have been used.
- We apologize that parts of the manuscript were not clear about which SWE scenario we used. For lines 240 to 276, the analysis around Figures 4 and 5 used the model results from the heterogeneous (i.e., "natural") snow distribution and the SWE average from the given CRNS footprint. We have revised these sections to increase clarity.

- The results and discussions around Figure 4 and 5 seem straight forward. However, it is questionable if the "snow-free" day is a good choice for an analysis of the effect of snow cover. If the SWE average is based on the CRNS footprint in this analysis, almost all virtual detector locations are compared under completely snow-free

conditions, except for the sensors close to the remaining snow patch ("snow drift"). Choosing a day with a more prominent snow cover (e.g. 17 February) would be more relevant.

- Results and discussions around Figure 6 and 7 would benefit from additional information on how much each virtual detector was affected by fractional snow cover throughout the study. This would strengthen the discussion, which seems to evaluate the complexity of snow cover within the footprint area from visual inspection.

- We are responding jointly to the two comments above, since they seem related. Our intent in Figures 4 and 5 was to illustrate the influence of spatially limited, high SWE snow drifts on our CRNS results. We felt that January 15th was ideal for this because of its lack of snow cover outside of the snow drift. The snow distribution from other dates would include this effect, but it would be overprinted by the influence of snowpack elsewhere in the CRNS footprint. We accept the criticism that this example doesn't necessarily show all of the considerations that influence the CRNS model results. The hard reality is that field data collection in this cold, windy environment is challenging, making these types of analyses and techniques even more valuable for better understanding snow water on the landscape.

  We also accept the feedback on Figures 6 and 7. We agree that the analysis can further benefit from how the virtual detector was affected by fractional snow cover throughout the study. We will add qualitative comparisons to our discussion. To that end, we have changed the order of the results presented in Section 4.1. We now present the complete results first (former Figures 6 and 7), which are all affected by the heterogeneous snow distribution, and add a color scale to the points to reflect the fractional snow cover. Then, we discuss how the snow drift also affects our results (former Figures 4 and 5).

- The analysis of section 4.2 and 4.3 give a great added value to the study. While results of 4.2 are partially mentioned in the abstract (l. 20-22) and a hint on 4.3 is provided in the introduction (l. 94-95) they appear hidden and should be more clearly visible, in both abstract and introduction.

- Thank you for this comment. We agree sections 4.2 and 4.3 are important to this study and should be highlighted in our abstract and introduction. We have edited our abstract and introduction to include these results.

- The analysis in 4.1 shows that CRNS measurements on the "snow-off" day (January 15) were affected by the snow drift, presumably lowering the $N\theta$ that was chosen for the SWE conversion. The effect on the converted SWE signal should be briefly discussed in 4.3.

- We agree and we have included a brief discussion of the converted SWE signal into 4.3. We have also revised Fig. 9 to include the SWE calculated using bare ground conditions as the baseline.

- Consider rephrasing line 480 to 482 for better logical reasoning and more clarity.
- Thank you for pointing out this lack of clarity. We have rephrased lines 480 to 482.

Illustration remarks
- Figure 1:
  - For clarity, the position and viewing direction of these images could be marked in Figure 2.
  - Thank you for this comment. We have added markers to Figure 2 that clarify the position and viewing direction of our images on Figure 1.

- Figure 3:
  - A different color should be applied to snow-free areas to allow for a differentiation into areas of heterogeneous snow cover and areas of partial snow cover.
  - We agree that marking the snow-free areas and areas of partial snow cover may be beneficial and clearer to readers. We have changed Figure 3 to include these no-snow masks (see example figures below, with no snow areas shown in gray). However, we must note that the snow was very shallow for many of our observation dates, and orthophotos were only available for one of the dates, so we cannot be completely certain about the fractional snow cover percentage across all dates. The uncertainty that exists with our lidar measurements were outlined in Woodley et al. (2024), with RMSE values between 4 and 7 cm. The high RMSE values were likely from the wheat stubble giving a false return. There is potential that an incorrect threshold snow depth for delineating snow-covered vs. snow free areas could drastically change the fraction of snow cover within the study area. However, we compared our masks (using 0 cm snow depth as "no snow") with a snow cover class analysis of the CARC conducted by Palomaki and Sproles (2023). We are including Figure 1d and 1e into this discussion from Palomaki and Sproles (2023) which shows that creating a snow cover mask using a threshold snow depth of 0 cm matches the snow cover class analysis from an orthomosaic photo on 21 Jan. We have included discussions of this uncertainty in our revised manuscript as well.

[Figure]

*Figure R1. Lidar snow depths (SD) in m for 21 January 2021. Snow free pixels are shown as grey. Snow free pixels are any pixels with a SD equal to 0 m.*

[Figure]

*Figure R2. Lidar snow depths (SD) in m for 21 January 2021. Snow free pixels are shown as grey. Snow free pixels are any pixels with a SD less than 4 cm (0.04 m). This threshold was chosen due to the uncertainty in the lidar flights.*

[Figure]

*Figure R3. Figure 1(d) from Palomaki and Sproles (2023). An orthomosaic image of the CARC on 21 January 2021 with a spatial resolution of 10 cm.*

[Figure]

*Figure R4. Figure 1(e) from Palomaki and Sproles (2023). The snow cover classes at the CARC at a spatial resolution of approximately 5 m.*

- The choice of an exponential color scale is reasonable, but should be better indicated in the legend (e.g. color bar with exponential color distribution, instead of even increments)
- We understand the reviewers point that showing the colorbar on an exponential scale would be a clear signal to the reader that the colorbar is not linear.  However, we found that the exponential scale makes the tick mark values harder to read, as it is harder to differentiate the colors when the ticks are compressed into the upper portion of the colorbar.  While the scale is nonlinear, we think that showing a set number of categories makes the snow depth more interpretable to the reader.   We have attached an example of the figures below. We found that the differences between the colormaps are very minor. However, we note that it is not possible to show a value of 0 with a log distribution, so we do lose any values between 0 and 1 cm (0.01 m). For these reasons, we have retained the current color scale on our figures. However, we now note the irregular color scale in the figure captions of the revised manuscript so that readers are aware.

[Figure]

*Figure R5. Lidar DSM of snow depths at the CARC for 21 January 2021. The colorbar is the same colorbar as the manuscript.*

[Figure]

*Figure R6. Lidar DSM of snow depths at the CARC for 21 January 2021. The colorbar distribution is now exponential.*

- o The images miss a scale bar. A dashed line that indicates the domain outline as in Figure 2 would be additionally interesting, as well as the distance of the outer virtual detector locations to the domain boundary.
- o We thank the reviewers for their comment. We have added a scale bar to the maps. To address the second part of the comment, all of these maps are within the dashed domain outline in Figure 2, which is why we did not plot it in Figure 3. The footprint for p04 was added to illustrate what the comment suggested. We understand that this was not clear. We have added a label to the x-axis on Fig. 3b like the ones in Figures R5 and R6 to show it was a 1000m and will clarify in the caption of Figure 3 that our study area was 1000 m by 1000 m.

- Figure 5:
    - o For consistency, the color scale in e) to f) should be the same as in the previous figures (white indicating low snow and blue indicating high snow accumulation). Further, the SD maps miss a scale bar.
    - o In the original Figure 5, we reversed the colormap because the SD maps would blend into white background of the figure but kept the colors consistent. We have altered the figure so that the colors are consistent. Also, we have added scale bars to our SD maps.

    - o Since the findings at P00 and P19 are contrary (larger changes on the snow side) to the findings at P07 and P05 (larger changes on the no-snow side) besides the similarity in snow distribution, P19 should also be presented in this figure.

- o We originally left off P10 from figure 5 because we felt the individual panels would have been too small to make out any details. We have included P19 in Figure 5 to highlight the contrary findings.

- Figure 6 & 7:
  - o The figure should indicate which scenarios were included in the analysis (all except 15 January).
  - o We apologize for the lack of clarity. All scenarios were used in this figure. We have clarified this in the text and caption.

  - o Coloring the scatter plot after the snow cover fraction within the corresponding virtual detector footprint may add valuable insights.
  - o We agree that coloring the scatter plot by the snow cover fraction may yield valuable insights. We have revised the figure accordingly.

- Figure 8:
  - o The choice of red as a color for agreement seems counter intuitive. Green may be a better choice (the significance of that color needs to be indicated in the legend).
  - o We understand the reviewer's comment about our choice of color, however, selecting a decent color to contrast with the blue is a challenge. Green may not be the best choice if we want to have a figure that is colorblind friendly. We can potentially change it to orange, but feel the red color serves the purpose of being visible and distinguishable from the background colormap.

  - o All maps miss a scale bar.
  - o We have added a scale bar to all figures that show maps.

  - o The exponential character of the SWE color bar should be displayed with exponential color increments.
  - o This comment is similar to a previous comment about exponential color increments. Please refer to our response above that includes Figures R5 and R6.

References:
Palomaki, R. T. and Sproles, E. A.: Assessment of L-band InSAR snow estimation techniques over a shallow, heterogeneous prairie snowpack, Remote Sensing of Environment, 296, 113744, https://doi.org/10.1016/j.rse.2023.113744, 2023.

---

## Referee Report (RR1)

**Review on "Influence of Snow Spatial Variability on Cosmic Ray Neutron SWE: Case Study in a Northern Prairie" by Kim et al.**

**Major applied changes**

- The order in which the URANOS derived results are presented has been reversed, which improves the clarity of the manuscript: (1) modelling all 8 snow scenarios, (2) focusing on 15th Jan. snow scenario, (3) focusing on points with highest difference in neutron counts (P07, P00, P05, P19).
- Removing the plot that showed the analysis of Figure 4 for average SWE of the model domain, and instead only describing the results of this analysis improves the structure of the manuscript.
- Chapter 4.4 on "Assumptions and Limitations of this Study" was added and provides a great reflectance on strengths and weaknesses of the presented study.
- The effect of Fractional Snow Cover is appropriately discussed and displayed.

**Minor comments**

- Line 10: "[..] noninvasive (or aboveground) […]"
  - Please consider, that the method is both, non-invasive and above ground.
- Line 24: "[…] while CRNS SWE values match more closely."
  - Please provide the reference (compared against what?).
- Line 67-71: "In addition, continuous SWE monitoring through snow pillows or snow scales like those found in the snow telemetry (SNOTEL) network from the US Department of Agriculture Natural Resources Conservation Service (USDA NRCS) (Serreze et al., 1999), are not as effective in the prairie due to wind erosion and transport."
  - This is a great argument to use CRNS. Maybe consider to mention this point in the discussion to support your point that CRNS is especially well suited in a prairie environment.
- Line 117: "[…] during DJF[…]"
  - Please specify DJF
- Line 177-181: "Thus, we utilized a 2-layer density scheme to calculate spatially distributed SWE at the CARC, using snow density values derived from the snow pit measurements. The thickness of the lighter and basal snow layers on a given date was determined by differencing the lidar DSMs on different dates. These 2-layer snow density and depth maps were used to specify the "natural" snow cover conditions in the neutron transport simulations (section 3.2)."
  - This description seems to indicate that you defined a stratified snow layer in URANOS consisting of a denser bottom layer and a less dense top layer when simulating the neutron interactions with the lidar derived heterogeneous SWE maps. In section 3.2 (line 236) you write, however, that you created "[…] a snow layer with uniform thickness and density". The latter phrase in section 3.2 may correspond only to the uniform SWE cases, such that the provided information is not contradicting. However, to make your model setup more clear, please describe in section 3.2 also how you set up the heterogeneous snow scenarios in URANOS.
- Line 233: "[…] snow water volume was divided"
  - Not clear. Maybe "was derived"? Please consider to reformulate.

- Line 370-375: "We noticed skews in neutron origins due to the relation of the model geometry, namely the position of the virtual detector and the source geometry. Virtual detectors placed closer to the edges of our domain had neutron origins that were skewed towards the center of the domain. Therefore, we limited the neutron counts to within a 200 m radius of the virtual detector."
  - Please consider to move this paragraph on how you have been dealing with model boundary artefacts to the methodology section (3.2), since this effect must have affected all URANOS simulation runs.
- Line 387-388: "P05, P07, and P19 which were modelled closer to the snow drift."
  - Incomplete sentence
- Line 390: "[…] which enhances the neutron counts on the snow side in the heterogeneous runs."
  - Is the figure not showing a decrease in neutron counts on the snow side (in line with what you wrote previously in this paragraph)?

**Illustration remarks**

- Figure 3:
  - Thank you for visually distinguishing snow-free from snow covered areas! This gives more in-depth information to the figure. Please consider to add an additional field to the legend, indicating that grey color corresponds to snow-free areas (instead of describing it only in the figure caption).
- Figure 5:
  - Please consider coloring the point plot in 5 (a) after bare ground (in the same fashion as in Figure 4 (a)), as this would make the influence of the distance of snow drifts to the detector location even better visible.
- Figure 4 (page 17):
  - The numbering of the figure should continue as Figure 7.
  - For c, d, e, and f: Please use the same color scale as in Figure 3 (where snow-free areas are indicated in grey).
- Figure 85 (page 19):
  - The numbering of the figure should continue as Figure 8.

---

## Author Response (AR3)

We thank Nora Krebs and Dr. Paul Schattan (hereafter "the reviewers") for their comments that improved the quality and clarity of our manuscript. Our responses to each comment are highlighted in blue. The line number used in the responses are the line numbers in the non-marked manuscript.

Review on "Influence of Snow Spatial Variability on Cosmic Ray Neutron SWE: Case Study in a Northern Prairie" by Kim et al.

Minor comments
- Line 10: "[..] noninvasive (or aboveground) [...]
    - "Please consider, that the method is both, non-invasive and above ground.
    - Thank you to the reviewers for pointing this out. We meant that noninvasive and aboveground were interchangeable, however, we apologize if that meaning was not clear as previously written. We have edited line 10 to read "[...] noninvasive, aboveground [...]".
- Line 24: "[...] while CRNS SWE values match more closely."
    - Please provide the reference (compared against what?).
    - Thank you to the reviewers for pointing out the lack of clarity. We have edited the sentence to read "CRNS showed better agreement with lidar-derived SWE at our prairie site compared to several gridded snow products."
- Line 67-71: "In addition, continuous SWE monitoring through snow pillows or snow scales like those found in the snow telemetry (SNOTEL) network from the US Department of Agriculture Natural Resources Conservation Service (USDA NRCS) (Serreze et al., 1999), are not as effective in the prairie due to wind erosion and transport."
    - This is a great argument to use CRNS. Maybe consider to mention this point in the discussion to support your point that CRNS is especially well suited in a prairie environment.
    - We thank the reviewers for the constructive feedback. We agree that CRNS is well suited for snow research in a prairie environment. We apologize if this statement was not made clear in our manuscript, since we felt our results and discussion in sections 4.2 and 4.3 were making these claims. We acknowledge that this claim may not have been made explicitly within our manuscript. Therefore, we have made minor edits to our manuscript so that this point is stated clearly. We have edited the sentence beginning on line 77 to clearly respond to the previous claims: "To overcome these limitations in snow observations in the prairies, [...]". Furthermore, we have edited the sentence starting on line 476: "These results indicate that CRNS provides

value for large-scale SWE estimates in the prairies, and well suited to measure SWE in prairie environments compared to the conventional, smaller-footprint sensors."

- Line 117: "[…] during DJF[…]"
  - Please specify DJF
  - We apologize for not specifying DJF. We have rewritten line 117 to "December-February (DJF)".
- Line 177-181: "Thus, we utilized a 2-layer density scheme to calculate spatially distributed SWE at the CARC, using snow density values derived from the snow pit measurements. The thickness of the lighter and basal snow layers on a given date was determined by differencing the lidar DSMs on different dates. These 2-layer snow density and depth maps were used to specify the "natural" snow cover conditions in the neutron transport simulations (section 3.2)."
  - This description seems to indicate that you defined a stratified snow layer in URANOS consisting of a denser bottom layer and a less dense top layer when simulating the neutron interactions with the lidar derived heterogeneous SWE maps. In section 3.2 (line 236) you write, however, that you created "[…] a snow layer with uniform thickness and density". The latter phrase in section 3.2 may correspond only to the uniform SWE cases, such that the provided information is not contradicting. However, to make your model setup more clear, please describe in section 3.2 also how you set up the heterogeneous snow scenarios in URANOS.
  - We apologize if this was not clear. Most of this work was built off of the modeling that was done in the Water Resources Research (WRR) article of Woodley et al. (2024). However, since not every reader will have not read this article, we agree that not clearly describing our methodology for the heterogeneous snow cover model runs would be confusing. As a result, we have added a brief summary by changing the sentence on line 204. The sentence now states: "Our "natural" or heterogeneous model setups are similar to the simulations described in Woodley et al. (2024), with a stratified 2-layer snow density model as described in Sect. 3.1 and split into semi-regular layers (see colorbar on Fig. 3)." In addition, we moved the first sentence of the following paragraph (line 228) into this paragraph to add more context in the beginning of our methodology to be clear about the three different snowpack scenarios.
- Line 233: "[…] snow water volume was divided"
  - Not clear. Maybe "was derived"? Please consider to reformulate.

- o We thank the reviewer for catching this lack of clarity. The sentence was reformulated to be clearer. It now reads: "We derived the uniform snowpack thickness by dividing the total amount of snow water volume by the snow density of hard coded material values of different snow types in URANOS."
- Line 370-375: "We noticed skews in neutron origins due to the relation of the model geometry, namely the position of the virtual detector and the source geometry. Virtual detectors placed closer to the edges of our domain had neutron origins that were skewed towards the center of the domain. Therefore, we limited the neutron counts to within a 200 m radius of the virtual detector."
    - o Please consider to move this paragraph on how you have been dealing with model boundary artefacts to the methodology section (3.2), since this effect must have affected all URANOS simulation runs.
    - o We have carefully considered this comment, but we have not moved this paragraph to our methodology section. The reason that we did not make the suggested change is that we did not apply the 200 m radius restriction to all of our model results in the manuscript, but instead only for the analysis shown in Fig. 6. Therefore, we felt that moving this paragraph might mislead readers to think that this method was applied to all of our model results.
    - o With regards to why the 200 m radius restriction was not applied to the rest of our analysis (e.g., the results shown in Fig. 4), the comparisons we make between the uniform and heterogeneous URANOS runs will be equally skewed based on the location of the virtual CRNS within the model domain. For example, for location P05 on 17 Feb. 2021 for uniform vs heterogeneous snowpacks, the location of the virtual detector will have the same effect on the neutron field for both scenarios, as both virtual detectors are situated in the same location within the model domain. However, the snow distribution does change, and thus the difference in the neutron counts between the two scenarios will reflect only the change in the snowpack. In Figure 6, we wanted to compare the neutron counts near the 171 m footprint of the CRNS and how the snow drift would have affected the neutron counts in its immediate surroundings where we know the CRNS is the most sensitive.
    - o We apologize if our methodology was not clear in the manuscript. We have edited the last sentence of this section to clearly state that we this analysis was applied to Figure 6 results only: "[...] within a 200 m radius of the virtual detector only for the results shown in Fig. 6(a)-6(d)".
- Line 387-388: "P05, P07, and P19 which were modelled closer to the snow drift."
    - o Incomplete sentence

- o Thank you to the reviewers for catching this sentence clause. We have incorporated this clause into the previous sentence. Additionally, to be clearer about our results and discussion, we have made changes to Lines 387-394.
- Line 390: "[...] which enhances the neutron counts on the snow side in the heterogeneous runs."
  - o Is the figure not showing a decrease in neutron counts on the snow side (in line with what you wrote previously in this paragraph)?
  - o Thank you to the reviewers for catching this mistake. We have changed the sentence to read: "[...] which reduced the neutron counts on the snow side."

Illustration remarks
- Figure 3:
  - o Thank you for visually distinguishing snow-free from snow covered areas! This gives more in-depth information to the figure. Please consider to add an additional field to the legend, indicating that grey color corresponds to snow-free areas (instead of describing it only in the figure caption).
  - o Thank you to the reviewers for pointing this out. We have added a label for the snow free areas to the legend of Figure 3.
- Figure 5:
  - o Please consider coloring the point plot in 5 (a) after bare ground (in the same fashion as in Figure 4 (a)), as this would make the influence of the distance of snow drifts to the detector location even better visible.
  - o Thank you for this suggestion. We have made this change and thank the reviewers for improving the quality of our figure. It adds more context to our results. Additionally, it allows us to make our unique points more distinct. We have also changed the markers for P05, P07, and P19 from circles to their own unique shape (square, hexagon, and diamond, respectively) to add additional context where each point is located. We have updated the caption to Fig. 5 to reflect the changes made to the figure.
- Figure 4 (page 17):
  - o The numbering of the figure should continue as Figure 7.
  - o Thank you for the reviewers for catching this typo. We have renumbered this figure and all figures to be numbered correctly.
  - o For c, d, e, and f: Please use the same color scale as in Figure 3 (where snow-free areas are indicated in grey).

- o We have carefully considered this comment, but ultimately chose not to make the suggested change to Figure 7.  We agree with the Reviewers that consistency in visualization throughout a manuscript is important.  However, Figure 7 is in some ways distinct from the previous figures in what it shows and in what we hope to convey. First, Figures 7c-7f are plotting different variables than Figure 3 (SWE instead of snow depth), and the SWE shown in Figure 7 is from synthetic calculations to compare what a CRNS or snow scale might measure, rather than an observed SWE distribution.  In this sense, the idea of "snow-free" areas becomes slightly abstract, especially for panels 7(c) and 7(d) that show synthetic CRNS SWE estimates (derived from spatial integration of lidar-derived SWE estimates). Most importantly, the goal of Figure 7 is to illustrate the difference in representativeness of a CRNS compared to a snow pillow, so the differences in the spatial coverage and distribution of the red areas between 7(c) and 7(e), and 7(d) and 7(f), are the most important takeaway for the reader.  We feel that adding gray to panels 7(e) and 7(f) but not panels 7(c) and 7(d) (because panels 7(c) and 7(d) do not have any "snow-free" areas, given the spatial weighting function of the synthetic CRNS SWE) would makes this comparison visually difficult.
- Figure 85 (page 19):
  - o The numbering of the figure should continue as Figure 8.
  - o Thank you for the reviewers for catching this typo. We have renumbered this figure and all figures to be numbered correctly.

Other minor changes:
- To be consistent, we have changed all instances of "modelled" to "modeled" in the "manuscript" to be consistent throughout the manuscript.
- To keep our abstract under 250 words, we have made a small edit on line 20 from "low amounts of SWE" to "low SWE".